# A simple model for Behavioral Time Scale Synaptic Plasticity (BTSP) provides content addressable memory with binary synapses and one-shot learning

Yujie Wu[1] & Wolfgang Maass [2] ✉

Recent experimental studies in the awake brain have identified a rule for synaptic plasticity that is instrumental for the instantaneous creation of memory traces in area CA1 of the mammalian brain: Behavioral Time scale Synaptic Plasticity. This one-shot learning rule differs in five essential aspects from previously considered plasticity mechanisms. We introduce a transparent model for the core function of this learning rule and establish a theory that enables a principled understanding of the system of memory traces that it creates. Theoretical predictions and numerical simulations show that our model is able to create a functionally powerful content-addressable memory without the need for high-resolution synaptic weights. Furthermore, it reproduces the repulsion effect of human memory, whereby traces for similar memory items are pulled apart to enable differential downstream processing. Altogether, our results create a link between synaptic plasticity in area CA1 of the hippocampus and its network function. They also provide a promising approach for implementing content-addressable memory with on-chip learning capability in highly energy-efficient crossbar arrays of memristors.

The brain has to solve a really challenging algorithmic problem when it stores episodic memories: It has to allocate on-the-fly neurons that form a trace or tag for a new experience that can subsequently be reactivated with partial cues. In other words, it needs a mechanism for creating a content-addressable memory (CAM) through one-shot learning. Recent experimental work has elucidated an essential ingredient of that: Behavioral time scale synaptic plasticity (BTSP)[1–3]. BTSP differs strongly from plasticity rules such as the Hebb rule and STDP (spike-timing-dependent plasticity) that have previously been considered in efforts to model the creation of memory traces in the brain. This is especially salient because in contrast to most experimental data on synaptic plasticity[4], BTSP has been validated through experiments in awake and behaving animals. Other converging experimental data suggest that area CA1 of the hippocampus, where BTSP has been demonstrated, is a brain area that is central for the creation of memory traces for episodic and conjunctive memories[5].

BTSP differs in the following aspects from traditionally studied plasticity rules:
- BTSP does not depend on the firing of the postsynaptic neuron.
- Instead, it is gated by synaptic input from another brain area, the entorhinal cortex (EC). These gating signals appear to be largely stochastic[6].
- BTSP does not require dozens of repetitions of a protocol for the induction of synaptic plasticity, but is effective in a single or few trials, i.e., it provides a mechanism for one-shot learning[1].
- The direction in which BTSP changes a synaptic weight depends primarily on the preceding weight value[3].

[1]Department of Computing, The Hong Kong Polytechnic University, Hong Kong, SAR, China. [2]Institute of Theoretical Computer Science, Graz University of Technology, Graz, Austria. ✉e-mail: maass@igi.tugraz.at

- BTSP acts on the time scale of seconds, a time scale that is suitable for creating episodic and other forms of conjunctive memories that integrate temporally dispersed information.

A mathematical model for BTSP and its impact on the induction of place cells was provided by ref. 3. There, and in the subsequent modelling studies[7,8] for the induction of place cells, BTSP was modelled through a differential equation with continuous time and weights but without noise. However, the experimental data for BTSP, see Fig. 3C in ref. 3, indicate a substantial amount of trial-to-trial variability of weight changes. Hence, the question arises whether the essence of experimentally observed BTSP can also be modelled by a simple stochastic rule for BTSP with binary weights.

The first rule for BTSP with binary weights had been proposed in a preliminary version[9] of this work. Subsequently, a similar rule was also proposed as a model for new experimental data for BTSP in area CA3[10]. We will examine here the system of memory traces in area CA1 that is created by the rule[9] and its stochastic variant. Since these rules were chosen to be simple, this analysis can be carried out not only through numerical simulations but also analytically. In this way, one can even analyze the resulting memory capacity at the scale of the brain.

We find that BTSP creates through one-shot learning a high-capacity CAM with just binary weights. This is remarkable since the most commonly studied CAM model, the Hopfield network (HFN), requires a number of weight values that grows linearly with the number of memory items (one often refers to this case as "continuous weights"). HFNs have much lower memory capacity when continuous weights are rounded to binary weights[11], and we are not aware of online learning rules for HFNs with binary weights.

The brain employs a sparse coding regime, where relatively few neurons are simultaneously active. This sparse coding regime contributes to its astoundingly low energy consumption and is, therefore, also desirable for neuromorphic hardware. Consequently, we focus our investigation on CAM for memory items that are represented through brain-like sparse activity, including the case of overlapping memory items.

Additionally, the human brain can not only create different memory traces for similar memory items, but memory traces for similar memory items exhibit a repulsion effect[12–14]. This effect, which could so far not be reproduced by HFNs or other learning rules, pulls memory traces for similar memory items actively apart, thereby enabling differential downstream processing. We show that BTSP can reproduce this repulsion effect of memory traces in the human brain.

An especially interesting facet of experimental data on BTSP is its dependence on stochastic input signals from area EC that cause plateau potentials in CA1 neurons[6]. We show through theoretical analysis and numerical simulations that the probability of the occurrence of such a plateau potential is a critical parameter for the performance of resulting memory systems. It has no analogue in previous memory models or rules for synaptic plasticity. Our analysis suggests that the experimentally observed value of this parameter is close to optimal for the quality of the memory system that BTSP creates, and that the several seconds long time window of plasticity that is opened by a plateau potential plays a central role in bringing it into this range.

The creation of memory traces through BTSP can be seen as an expansion of another well-known biological algorithm that also relies on stochasticity, the fly algorithm[15]. The fly algorithm also employs binary weights, but these are assumed to exist a-priori; they are not learnt. In contrast, BTSP produces a random projection from input patterns to memory traces through learning. In other words, the random projection that BTSP creates is custom-made for a particular ensemble of input patterns and can be continually extended through one-shot learning. We demonstrate that this difference from the fly algorithm, and from random projections in general, enables BTSP to create attractors around stored memory traces that support stable

recall from partial cues. In addition, BTSP manages to overcome two bottlenecks that the fly algorithm faces when one wants to apply it to larger neural systems: It neither requires a projection of input patterns into a 40 times larger neural population, nor a global winner-take-all competition over the population of neurons into which input patterns are projected. These two requirements are met by the relatively small olfactory system of the fly but not by the memory system of the mammalian brain. The number of neurons in area CA1, the area into which memory items are projected from area CA3, is not 40 times larger but at most 1.5 times larger than area CA3, both in the rodent and in the human brain (see Table 3-1[16]). Furthermore, area CA1 consists of 0.39 to 14 million neurons in the mammalian brain[16], and the assumption of a winner-take-all computation is more problematic for such a large system. Hence, evolution had to invent a different method for storing large numbers of memory traces in the mammalian hippocampus: BTSP.

Since BTSP is a local synaptic plasticity rule that requires in its simplified form only two weight values, it is especially suited for on-chip learning in neuromorphic hardware, in particular for online creation and expansion of CAM in crossbar arrays of memristors with just two required resistance states. Hence, this BTSP-based CAM paradigm provides a a new approach for creating highly energy-efficient large-scale implementations of CAMs for brain-like sparse patterns. This approach can drastically expand the memory capacity of already existing implementations of CAM in crossbar arrays of memristors that were based on the HFN paradigm[17–19].

We review experimental data on BTSP in the next section and show that a simple plasticity rule with binary weights provides a good approximation to these data for the case that the arrival times of synaptic inputs are statistically independent from the arrival times of plateau potentials. We then address the main new parameter of this rule, the probability of a stochastic gating signal, and elucidate its functional impact. In the subsequent section, we compare the properties of the system of memory traces created through BTSP with the memory system created through a random projection, which is the main component of the fly algorithm. Subsequently, we present theory and numerical simulation results for the recall of memory traces for overlapping memory items for BTSP. We then compare the properties of the CAMs that are created through BTSP with feedback connections with properties of CAMs that are implemented by HFNs, both for the case of binary and continuous-valued synaptic weights. Finally, we show that BTSP reproduces the repulsion effect of the human brain and elucidate parameters of BTSP that are critical for this.

## Results

### A simple rule for synaptic plasticity captures core features of BTSP

We first recapitulate experimental data on BTSP, and then discuss options for modelling it using simple rules. Memory tags, also referred to as memory traces, are created in a few or even a single trial through the plasticity of synaptic connections from pyramidal cells in areas CA3 to pyramidal cells in area CA1 of the hippocampus. Figure 1A provides a schematic drawing of a pyramidal cell in area CA1, which receives synaptic input from area CA3, as well as synaptic input from layer 3 of EC on distal dendrites, which create plateau potentials in its apical tuft. Experimental data[1,6,20] show that these plateau potentials gate the plasticity of weights of synaptic connections from CA3 to pyramidal cells in area CA1. Since these plateau potentials and their impact on synaptic plasticity last for several seconds, the resulting plasticity process is termed behavioral time scale synaptic plasticity. But BTSP is not just a variant of STDP on a longer time scale. Unlike STDP, this synaptic plasticity mechanism does not depend on the relative timing of pre- and postsynaptic firing. Instead, it depends on the relative timing of presynaptic activity and a gating signal from EC3 in the postsynaptic neuron, whose arrival times are in first approximation

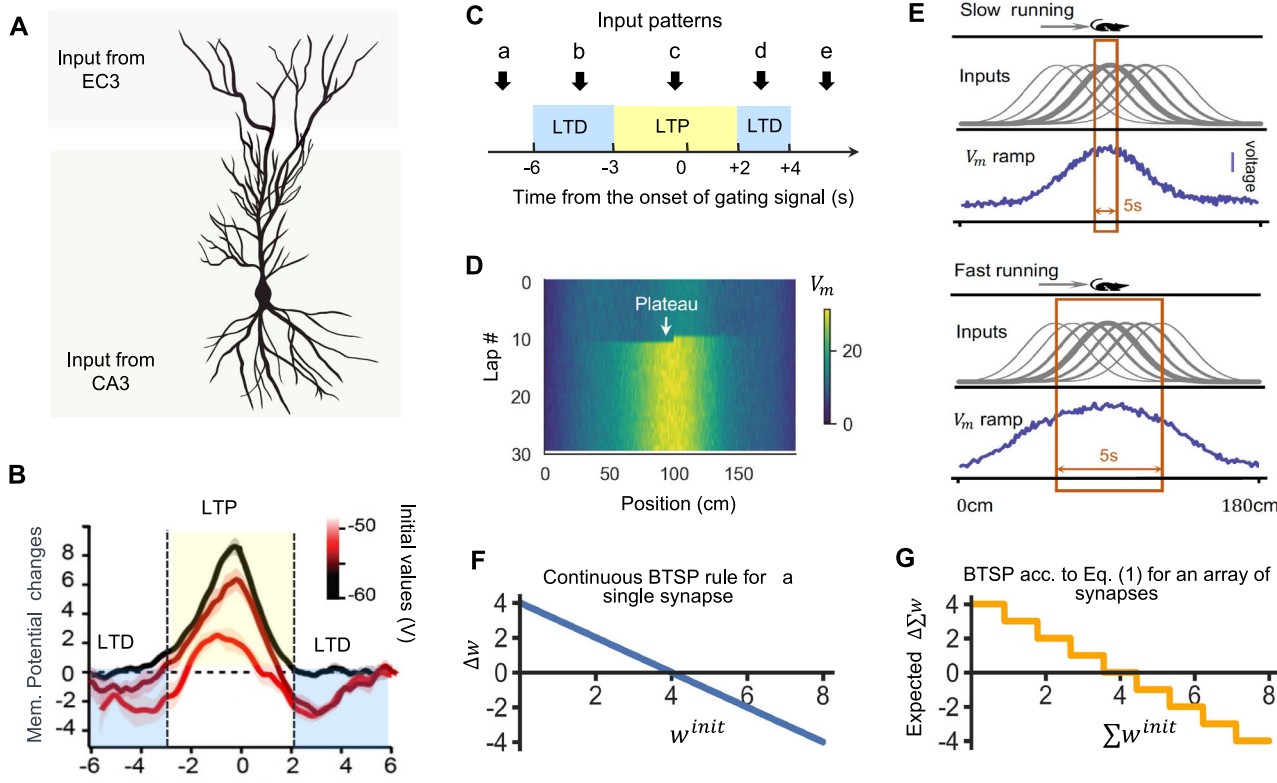

**Fig. 1 | Experimental data and a simplified rule for BTSP. A** A CA1 pyramidal cell receives synaptic inputs from at least two distinct sources: distal apical dendrites are activated by input from the entorhinal cortex (EC3), inducing plateau potentials[6], while proximal synapses are activated by input from CA3[16]. **B** According to the experimental data shown in Fig. 3E of ref. 3 synaptic weight changes through BTSP depend on the initial weight and the time of presynaptic firing relative to the onset of the gating signal (plateau potential). Changes are shown for three initial weight ranges (black: minimal, orange: maximal, dark red: intermediate), measured by the impact on the postsynaptic membrane potential $V_m$ The panel is redrawn from ref. 3. **C** The resulting scheme for the decision between long-term potentiation (LTP) and long-term depression (LTD). According to Fig. 3E[3] only input patterns that arrive up to 6s before or at most 4s after the initiation of a plateau

potential in the CA1 neuron are subject to synaptic plasticity through BTSP. Hence, patterns b, c, and d undergo plasticity, whereas patterns a and e do not. (**D**) Simulations using the simplified BTSP rule reproduce the emergence of new place fields, as seen in experimental data in Fig. 1A of ref. 1. **E** The simplified rule replicates the dependence of the spatial extension of place fields on the running speed, which demonstrates the seconds-long plasticity window of BTSP, see Fig. 1F in ref. 1. **F** Weight changes in dependence of the initial weight for a more complex biophysical model of BTSP according to Eq. (1) in ref. 3, where the initial weight can assume continuous values. **G** The simplified BTSP rule produces net changes in synaptic strength comparable to the biophysical model from panel E in the case of multiple (here 8) synaptic release sites, each with binary weights subject to Eq. (1).

generated through an independent and time-invariant stochastic process[6]. Hence, also the decision between LTP and LTD, that depends on this relative timing, is in first approximation stochastic. Furthermore, plasticity of synaptic weights is induced very fast, even in a single trial, but depends strongly on the current value of the synaptic weight[3].

The arrival of input from EC creates a plateau potential in the postsynaptic neuron in area CA1 that opens the plasticity window for several seconds before and after its onset (= time 0 in Fig. 1B). Experimental data on the resulting weight change exhibit substantial trial-to-trial variability, see Fig. 3B in ref. 3. But according to Fig. 1B, which is redrawn after Fig. 3E of ref. 3, the decision between LTP (long-term potentiation), LTD (long-term depression), and no weight change, depends on average on two factors: the prior value of the synaptic weight and the time difference of presynaptic firing and the onset of the plateau potential. Importantly, in contrast to Hebbian plasticity and STDP, BTSP does not depend on the firing of the CA1 neuron. The black curve in Fig. 1B shows that LTP occurs for synapses whose prior strength was small, provided that the synaptic input arrives within the interval from -3 to 2s relative to the onset of the plateau potential. Outside of this time interval, the weights of these synapses remain largely unchanged. The orange curve shows that LTD is induced for strong synapses, provided that the synaptic input arrives

within the windows from -6 to -3s or from 2 to 4s relative to the onset of the plateau potential. If the synaptic input arrives in between these time windows, the resulting weight change is negligible. Thus, the dependence of the decision between LTP and LTD on the relative timing between the arrival of the synaptic input and the onset of the plateau potential can be summarized by the simple scheme of Fig. 1C. Based on the assumption that the arrival times of synaptic inputs and plateau potentials are statistically independent, the chance that a synaptic input that arrives within the 10s long window of plasticity marked by the onset of a plateau potential falls into the LTP window is about 0.5, and the chance that it falls into the LTD window is also 0.5.

The dark red 3rd curve in Fig. 1B shows weight changes for the case of intermediate initial weight values, indicating a similar dependence of the decision between LTP and LTD on relative timing of synaptic input and plateau potential. But for the sake of simplicity and analytical traceability, we focus here, as in the BTSP model for CA3[10], on a simplified BTSP rule with binary synaptic weight values 0 and 1, that models the black and orange curves of the experimental data in Fig. 1B.

We approximate these data by the following simple rule: When an input pattern $x$ arrives within the time interval from -6 to 4s relative to the onset of a plateau potential in a neuron, its synaptic weights are increased with probability 0.5 (provided they are not yet at their

maximal value of 1), and also decreased with probability 0.5 (provided they are not yet at their minimal value of 0). This probabilistic decision depends on the relative timing of presynaptic firing and the stochastic onset of a plateau potential. Hence, the following rule is applied with probability 0.5 whenever synaptic input arrives within the 10s window marked by a plateau potential in the neuron:

$$w_i = \begin{cases} +1 & \text{if } x_i = 1 \text{ and } w_i = 0, \\ -1 & \text{if } x_i = 1 \text{ and } w_i = 1. \end{cases} \tag{1}$$

Otherwise, a synaptic weight remains unchanged (see section 2 of "*Methods*" for a more detailed justification).

We will refer in the following to this rule from Eq. (1) simply as BTSP rule. This rule is for the case of randomly occurring plateau potentials mathematically equivalent to the BTSP rule proposed for area CA3 in ref. 10, except that the relevant time windows have a slightly different length (see Section S1 of the Supplement). We show in Fig. 1D that one can reproduce with this rule the fundamental BTSP result from Fig. 1A of ref. 1: the instantaneous creation of a new place field when a plateau potential occurs while a mouse runs along a linear track (see Section S2 of the Supplement for details). In order to demonstrate a functional impact of the seconds-long plasticity window of BTSP, Bittner K C et al.[1] examined in their Fig. 1F the dependence of the spatial extension of this new place field on the running speed of the mouse. They reasoned that this spatial extension would be largely independent of the running speed in the case of a short plasticity window (on the scale of milliseconds or tens of milliseconds, see upper rows of Fig. 1E), but grow substantially with the running speed in the case of a seconds-long plasticity window. We show in Fig. 1E (lower rows) that our simple BTSP rule reproduces this characteristic fingerprint of a longer plasticity window in the resulting membrane potential of CA1 neurons after BTSP. Inputs to the model CA1 neuron are from model CA3 neurons whose diverse place fields are indicated by arrays of Gaussians. This demonstrates that, in contrast to STDP or most other rules for synaptic plasticity, the BTSP rule of Eq. (1) is a plasticity rule for the behavioral time scale of seconds (see section S2 of the Supplement for details).

One can further simplify this rule by deleting the dependence of weight updates on a stochastic events that occur with probability 0.5, and simultaneously reducing the probability of a plateau potential by this factor 0.5. Then, the effective learning rate remains the same. We refer to this variant of Eq. (1) as the core BTSP rule. Note that core BTSP is still a probabilistic rule because its weight updates depend on randomly generated plateau signals. We show in Fig. S1 that core BTSP yields virtually the same results as the rule from Eq. (1). However, we will base our theoretical analysis and numerical experiments on the stochastic rule from Eq. (1).

We demonstrate in Fig. 1F and G that the impact of a more complex rule for BTSP with continuous weight values, such as the rule proposed in ref. 3, can also be modelled by the simpler rule from Eq. (1) with binary synaptic weights. This holds if the pre-and postsynaptic neurons are connected by several synaptic release sites, whose individual strength can be approximated by binary values. To the best of our knowledge, the average number of release sites at a CA1 synapse has remained unknown. Hence, it remains open whether BTSP operates with continuously valued synaptic strengths or with a small number of discrete or even binary synaptic strengths at each synaptic release site. Our BTSP rule with binary weights has the advantage that it is theoretically better tractable. Also, neuromorphic implementations of binary weights through resistance values of memristors require substantially less technological effort than memristors that have to assume many different distinguishable resistance values[21]. Furthermore, the maximal number of distinguishable values is limited so that only HFNs with fewer than 1000 stored patterns can currently be emulated through memristor arrays.

## Setup for evaluating the creation of memory traces through BTSP

Converging experimental data, see ref. 5 for a review, suggest that memory traces for experiences (episodes) are created in area CA1 of the hippocampus through the application of BTSP to synaptic connections from area CA3 to CA1. Hence the input neurons of our simple network model shown in Fig. 2A correspond to pyramidal cells in the area CA3, and memory neurons to pyramidal cells in the area CA1. Since lateral connections between pyramidal cells in the area CA1 are rather rare, we model these memory neurons by a pool of $n$ disconnected McCulloch-Pitts neurons. In other words, each memory neuron compares its weighted sum of synaptic inputs from the input neurons to a firing threshold. It outputs 1 (or "fires") if the sum exceeds the threshold; otherwise, it outputs 0.

We consider the benchmark task to create and recall memory traces for $M$ input patterns $\boldsymbol{x}$, to which we will also refer as memory items in the following. Each input pattern is represented through the activity of $m$ input neurons; see Fig. 2A. For simplicity, we assume that their activity can be characterized as firing or non-firing, i.e., by a single bit, so that each input pattern can be described by a bit vector of length $m$. Although the arrival times of different components of the vector $\boldsymbol{x}$ may be dispersed over 100's of milliseconds or even seconds in the brain, plateau potential opens the window of plasticity in CA1 neurons for several seconds. This mechanism allows the integration of temporally dispersed patterns into a static memory trace, represented by the firing or non-firing of CA1 neurons in response to this input pattern. Importantly, each of these memory items $\boldsymbol{x}$ is presented just once to the network, while BTSP is applied to the synapses from the pool of $m$ input neurons to the pool of $n$ memory neurons. Performance for the benchmark task depends on various hyperparameters of the model. We focus on default values that are suggested by experimental data.

The parameter $m$ corresponds to the number of pyramidal cells in area CA3, which is estimated to be around 0.25 million in the rat brain, and around 2.83 million in the human brain. The parameter $n$ corresponds to the number of pyramidal cells in area CA1, which is estimated to be around 0.39 million in the rat brain, and around 14 million in the human brain. Since the large size of these values makes numerical simulations difficult, we have focused on values that are down-scaled by a factor of 1/10 from the size of these structures in the rat brain, and by a factor of 1/100 from the sizes in the human brain. Hence, our default values in the model are $m = 25,000$ and $n = 39,000$.

Pyramidal cells of area CA1 are estimated to have around 30,000 dendritic spines in the mouse, see chapter 5[16]. Since some presynaptic neurons are likely to have synaptic connections on several spines, we assume in our model that a memory receives on average synaptic inputs from a random set of 15,000 input neurons. This suggests a connection probability of $f_w = 0.6$ for the randomly chosen connections from input neurons to memory neurons. The fraction $f_p$ of bits that assume value 1 in an input pattern (corresponding to the fraction of neurons in CA3 that fire during the presentation of a memory item) was estimated by ref. 22 to have the value $f_p = 0.005$. Almost the same sparseness level was measured in a quite different brain area: Natural images were found to be encoded through the firing of a little less than 0.5% of neurons in the primary visual cortex of the monkey[23]. Obviously, neural codes with such a high level of sparseness are also essential for achieving brain-like energy efficiency in neuromorphic hardware. Hence, we use $f_p = 0.005$ as the default in our model for the fraction of 1's in a generic memory item. Note that in contrast to these biological data, most previous studies of memory models focused on neural codes for memory items with a much higher value of $f_p$. Since we are not aware of reliable experimental data on the firing threshold of pyramidal cells in CA1, we used grid search to set the firing thresholds of CA1 neurons at a value that optimally supports recall from partial cues (see *Methods*).

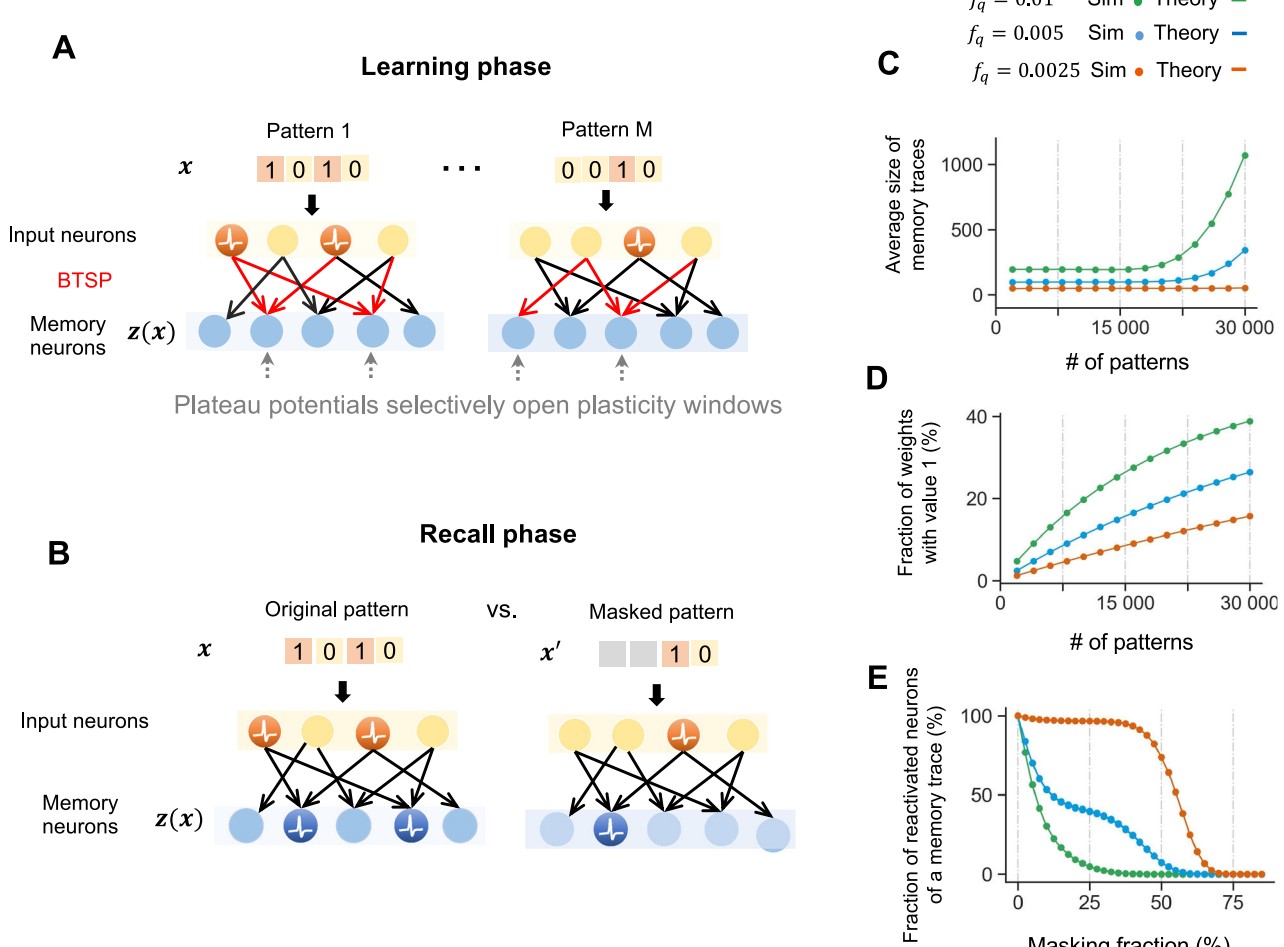

**Fig. 2 | Structure and functional properties of memory traces created through BTSP. A** Schematic illustration of the model for the creation of memory traces in the pool of memory neurons through BTSP. During the learning phase, synaptic connections between input neurons and memory neurons are subject to BTSP for a sequence of $M$ memory items, with each item being presented once (one-shot learning). For each memory item, a random set of memory neurons receives a plateau potential, which opens for this memory item the plasticity window for synaptic connections from active input neurons to this memory neuron. **B** During the recall phase, when no further learning takes place, partially masked versions $x'$ of previously learned memory items $x$ are presented, where a random fraction of 1's in the input pattern is replaced by 0's. **C** The average size of memory traces (= number of memory neurons that fire during recall with the full memory

item $x$ as cue) depends somewhat on the total number $M$ of memory items that have been learnt. It also depends on the probability $f_q$ of a plateau potential in a memory neuron during learning. In particular, values of $f_q$ that are substantially larger than the experimentally found value 0.005 lead to strongly increasing sizes of memory traces when many memory items are to be learnt. **D** The parameter $f_q$ also impacts the fraction of strong synaptic weights after learning, especially for large numbers of memory items. **E** A larger value of $f_q$ reduces the fraction of the memory trace that gets reactivated by a partial cue. One sees that the experimentally reported value 0.005 supports recall up to a fraction of one-third masked 1's. Each of the panels (**C**, **D** and **E**) show that the results of numerical simulations agree very well with theoretical prediction. The latter were obtained from Eqs. (13), (16) and (20).

After processing a sequence of $M$ memory items $x$ with one-shot learning through BTSP, a memory trace $z(x)$ emerges for each memory item $x$. It is defined by the set of memory neurons that fire after learning when input pattern $x$ is presented. A key question is what happens after learning when a perturbed or partially masked version $x'$ of one of the memory items $x$ is presented to the network, while BTSP is no longer active, see Fig. 2B. Here, we refer to the fraction of 1's in an input pattern that is replaced by 0 as the masking fraction. Recall with such masked cues is motivated by the goal of recalling conjunctive memories or episodes from just some of their components. We also considered recall with noisy perturbations of memory items where not only 1's are replaced by 0's but also 0's are replaced by 1's, but so that the expected number of 1's remains the same as in the original input patterns. We show in the Supplement that recall with such 'two-sided' errors in the cue is in general equally successful as recall with 'one-sided' errors in the cue, i.e., where only 1's are changed into 0's.

It is desirable that the set $z(x')$ of memory neurons that are activated by a masked or perturbed cue $x'$ is similar to the memory trace $z(x)$ for the original memory item $x$. Similarly as in ref. 24, we used the Hamming distance (HD) to measure their dissimilarity, as it is arguably the simplest and most direct measure of distance between binary vectors.

We also scale this HD by relating it to the average HD between memory traces for different memory items $x_a$ and $x_b$ from the original set of $M$ memory items. We refer to the quotient of these two HD's as relative dissimilarity. The memory trace $z(x')$ for a perturbed or masked input pattern $x'$ can only be seen as being similar to the memory trace $z(x)$ for the unperturbed memory item $x$ if this relative dissimilarity is substantially smaller than 0.5.

Note that both HD values that occur in this measure need to be analyzed after all input patterns in the sequence have been learned. This is essential since these HD values change in general during the processing of the sequence of the $M$ memory items through BTSP. In

particular, one needs to take into account that weights which have been increased when a memory item $x$ was processed through BTSP may become later subject to LTD when a subsequent item in the list of $M$ memory items has common 1's and also activates some of the same memory neurons, provided that these receive again a plateau potential when the subsequent memory item is processed. Therefore, one cannot use theoretical methods that have previously been developed for analyzing memory traces in HFNs, such as those in refs. [25–27], where the order of memory items during learning was irrelevant because the weight change did not depend on the prior value of the weight. Therefore, new tools from combinatorial mathematics and probability theory were needed to derive a theory for memory systems created by BTSP. The curves in Fig. 2C–E show that the resulting theory is very successful in predicting the outcome of numerical simulations.

## Functional role of the probability $f_q$ of the stochastic gating signal

Virtually all models for memory in neural networks are based on the postulate of Hebb: Synapses between neurons that fire together are strengthened. BTSP provides a radical departure from this postulate and also from STDP since BTSP does not depend on the firing of the postsynaptic neuron. Instead, it depends on presynaptic activity, the current value of the synaptic weight, and the current presence or absence of a plateau potential in the postsynaptic neuron. We later demonstrate the advantages of this mechanism in preserving previously formed memory traces during new learning and in facilitating the separation of similar memory items through a repulsion effect.

The experimental data[6] suggest that one can model the presence of plateau potentials in CA1 neurons by a stochastic rule, whereby CA1 neurons receive during learning of a new input pattern $x$ independently with probability $f_q$ a plateau potential. We show that the value of this parameter $f_q$, which has no analogue in previous memory models, significantly influences the properties of the resulting memory system. Figure 2C and D indicate that a larger value of $f_q$ increases the size of memory traces, i.e., the number of memory neurons that fire when a memory item is presented. However, it also increases the overlap of memory traces for different memory items, which is in general disadvantageous. Furthermore, Fig. 2E shows that a larger value 0.01 of $f_q$ reduces the capability to recall a memory trace with a partial memory item where a fraction of the 1's in the input vector have been masked, i.e., set to 0. This is also partially caused by the fact that plateau potentials not only enable LTP, but also LTD; the latter in case that the current weight is already large. Therefore, a larger value of $f_q$ implies that more weights that were first set to 1 for one item in the list of $M$ memory items become subject to LTD during the application of BTSP to subsequent memory items in the list. This tends to reduce the weighted sum that arrives at each neuron in the memory trace of a memory item. If in addition some 1's in the input pattern are set to 0, the resulting weighted sum may no longer reach the firing threshold of the memory neuron. On the other hand, Fig. 2E also shows that the relatively low value 0.005 for $f_q$ that is suggested by the experimental data[6] produces substantially more robust memory traces that support reliable recall even when a third of the bits of the memory item have been masked. We will later demonstrate that an even lower value of $f_q$ provides further performance improvement for recall with partial cues and input completion. On the other hand, we will also show that a lower value of $f_q$ reduces the repulsion effect.

Interestingly, the experimentally found seconds-long plasticity window of BTSP turns out to be essential for bringing the value of $f_q$ into the range around 0.005 that can be seen as a sweet spot with regard to the previously discussed functional impact of this parameter on the resulting system of memory traces. According to Fig. 4g of ref. 6, the rate of generation of plateau potentials in a CA1 neuron is around 0.0005 per second. But since the plasticity window that it

opens has a duration of 10s, presynaptic input that arrives at a random time point has a probability of 0.005 of falling into a plasticity window.

## BTSP creates a system of memory traces for sparsely encoded memory items that supports recall with partial cues

The fact that BTSP is gated by a stochastic component, random plateau potentials, endows the memory systems that it produces with generic advantages of stochastic memory allocation: The memory space, i.e., the population of memory neurons, is used uniformly, and artifacts that might result from a particular structure of the ensemble of input patterns are reduced. Random projection and random hashing are commonly used strategies for stochastic memory allocation in digital computers, see ref. 15 for a review. A random projection (RP) is a linear map from input neurons to memory neurons that is defined by a stochastic matrix of 0's and 1's, with subsequent thresholding of output values. In order to facilitate comparison, we use here the same fraction of 1's in the matrix of RPs that result from learning in the weight matrix for synaptic connections from input neurons to memory neurons in the case of BTSP.

A key difference between BTSP and RPs is that the projecting matrix for RPs is chosen independently from the memory items that are to be stored, whereas BTSP generates synaptic weights for a specific set of memory items. We demonstrate in Fig. 3 that this provides a substantial functional advantage to BTSP with regard to the recall of memory items with masked cues. The reason for this functional advantage is elucidated in panels A and B of Fig. 3. No matter how one chooses the threshold for binarizing the outputs of a RP, if one wants to have the weighted sum for a given number of memory items above this threshold, the weighted sum will be for many of these memory items just above the threshold, see Fig. 3A. Hence, masking even a small fraction of 1's in these memory items lets the weighted sum drop below the threshold, i.e., the memory neuron is not activated by the masked input pattern.

In contrast, the distribution of weighted sums that result from BTSP in a generic memory neuron is bi-modal, as shown in Fig. 3B: One cluster consists of weighted sums for memory items for which a plateau potential occurred in this neuron when the item was learnt, and the other cluster consists of memory items for which this did not happen. Further examples of this bimodal distribution are shown in Fig. S2 for other memory neurons. An obvious advantage of this bimodal distribution is that one can set the firing threshold of a memory neuron between these two clusters. Then, a perturbation of a small fraction of bits of a memory item is not likely to move the weighted sum to the other side of the threshold. This is confirmed by an evaluation for large sets of memory items in Fig. 3C and D: Recall with cues where up to a third of the input bits are masked works well for memory traces that are created by BTSP, but RP tolerates hardly any masking of input bits (even though we optimized its threshold for producing binary outputs for recall with masked cues, see "Methods"). Furthermore, it is shown in panel A of Fig. S3 that random perturbations of cues where also 0's are replaced by 1's is tolerated about as well as masked cues. Results of numerical simulations for the case of masked cues are corroborated by theoretical predictions that are depicted in Fig. S3. Simultaneously, one sees that the results of numerical simulations and theoretical predictions agree very well. We also found that BTSP creates at least equally good memory systems if one increases the probability of LTD and decreases the probability of LTP in Eq. 1, see Figs. S4 and S5.

HFNs are the most commonly considered CAM model. However, they require weights that can assume a large number of different values. The standard learning rule for storing memory items in a HFN[25,28] requires that the number of weight values grows linear with the number of stored memory items. We are not aware of other plasticity rules that install a functionally powerful CAM in HFNs with a much smaller number of weight values. In particular, there is

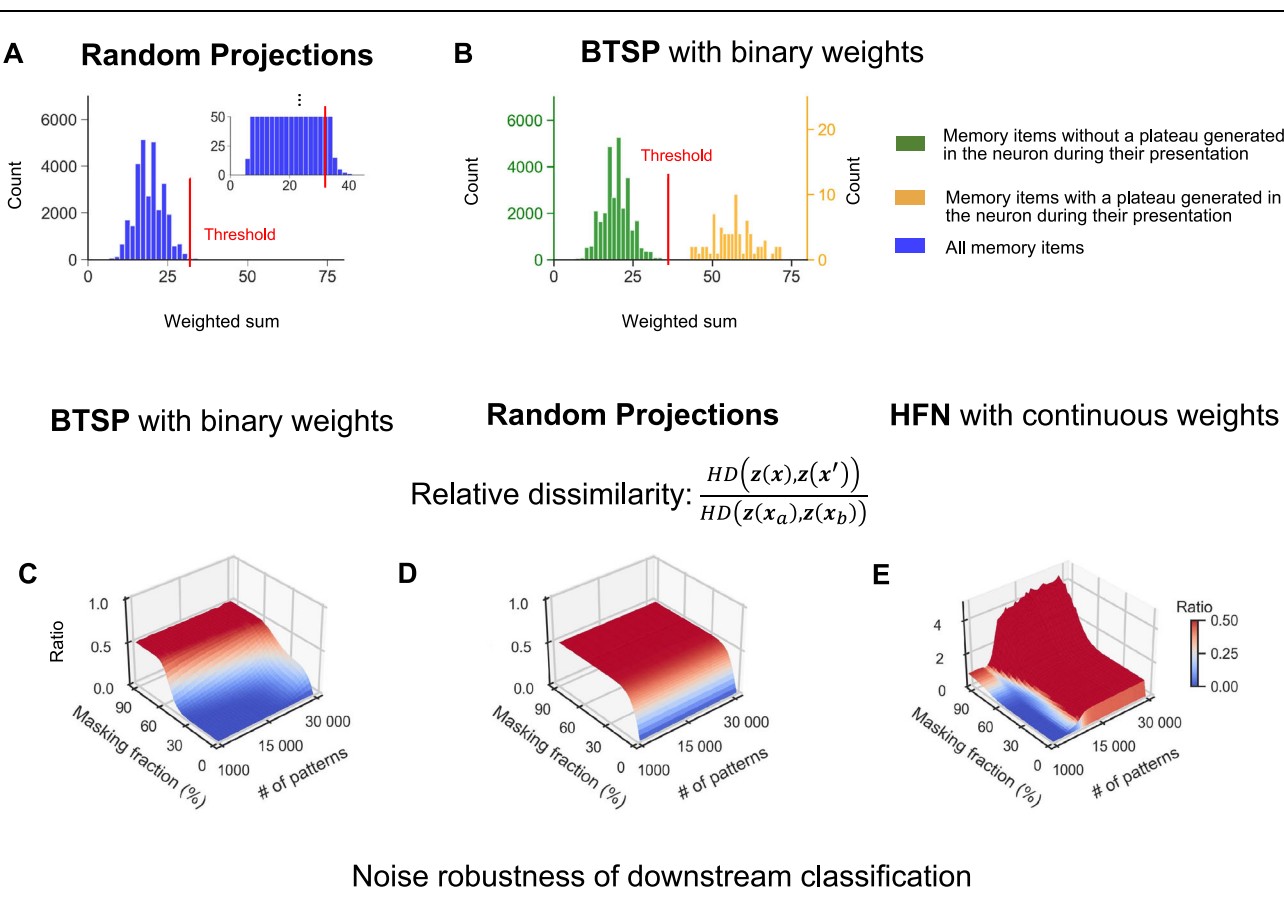

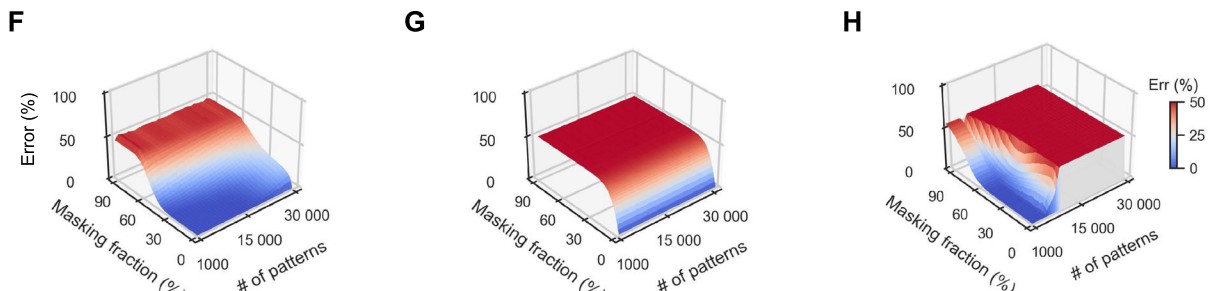

**Fig. 3 | Recall with partial cues for memory traces created via BTSP, a random projection (RP), and in a classical Hopfield network (HFN). A** The distribution of synaptic input sums to a sample memory neuron for all 30,000 memory items in the case of a RP. The red line (drawn for a refined scale in the inset) marks the optimal value of the neuron threshold for minimizing the relative dissimilarity for a masking fraction of 33%. Moving the threshold to a lower value would result in the inclusion of this neuron in a significantly larger number of memory traces, thereby increasing their overlap. **B** Distribution of synaptic input sums to a sample memory neuron for all 30,000 memory items that occur during learning with BTSP. Memory items associated with plateau potentials (yellow histogram) yield significantly larger sums than others (green histogram), allowing a threshold that separates these clusters. This creates a distinct memory trace in neurons that received plateau potentials, which is highly robust to the masking of cues. **C** and **D** show that the relative dissimilarity between the original memory trace $z(x)$ and the memory trace $z(x')$ recalled by a partial cue $x'$ is significantly smaller for BTSP than for a RP, due to the mechanism explained in panels A and B. With BTSP, dissimilarity remains low at masking fractions up to 33%, while RP quickly reaches a value of 0.5, where one can no longer distinguish to which memory item the trace for the masked cue belongs. **E** shows corresponding results for a classical HFN. Despite the substantially larger information content of its continuous weights, the memory capacity is not significantly larger than that of the memory systems with binary synaptic weights created by BTSP. **F**, **G** and **H** exhibit similar performance differences of BTSP, RP, and HFNs for generic downstream classifications of memory traces. Shown are simulation results averaged over 5 runs, with new random generation of memory items and plateau potentials.

apparently no learning rule that installs a powerful CAM in HFNs with binary weights. Furthermore, the results[11] show that if one first trains a HFN with a learning rule that employs continuous weights, such as the rule from ref. [29], and then quantizes the synaptic weights, the memory capacity of the HFN drops drastically.

Figure 3C and E show that the performance of memory systems with binary weights that are created through BTSP is close, and in some cases even superior to that HFNs with continuous weights. This result

is surprising because the continuous-valued weights of HFNs can store substantially more information than the same number of binary weights in the memory systems created through BTSP. Another functionally relevant difference is that the memory traces $z(x)$ for an input pattern $x$ is produced instantly by the BTSP network, but requires convergence to an attractor of the HFN. This usually takes a substantial number of time steps (we used 100 time steps in our simulations of HFNs). It is important to note that we considered HFNs that have the

**Simulation:**
**BTSP** with binary weights

**Theory:**
**BTSP** with binary weights

Relative dissimilarity: $\dfrac{HD\big(\boldsymbol{z}(\boldsymbol{x}),\boldsymbol{z}(\boldsymbol{x}')\big)}{HD\big(\boldsymbol{z}(\boldsymbol{x}_a),\boldsymbol{z}(\boldsymbol{x}_b)\big)}$

**A**

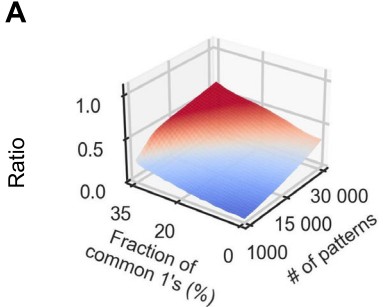

**B**

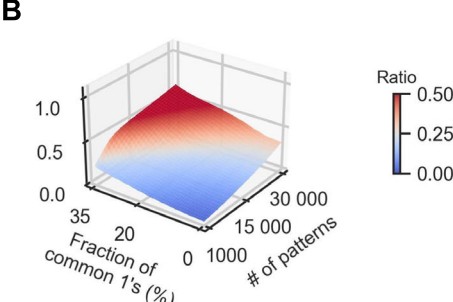

**Fig. 4 | Storing and recalling overlapping memory items.** Relative dissimilarity of memory traces for overlapping memory items, analysed through numerical simulations (**A**) and theory (**B**) for BTSP. The partial cues $\boldsymbol{x}'$ in the relative dissimilarity measure refer to patterns with 33% of elements masked. The relative dissimilarity is measured by the ratio $HD(\boldsymbol{z}(\boldsymbol{x}), \boldsymbol{z}(\boldsymbol{x}'))/HD(\boldsymbol{z}(\boldsymbol{x}_a), \boldsymbol{z}(\boldsymbol{x}_b))$, where $HD(\boldsymbol{z}(\boldsymbol{x}_a), \boldsymbol{z}(\boldsymbol{x}_b))$ refers to the average Hamming distance between the memory traces for any two different patterns $\boldsymbol{x}_a$ and $\boldsymbol{x}_b$ from the original set of memory items. A comparison of panels (**A** and **B**) shows that the theory for BTSP predicts the results of numerical simulations very well. The average results of 5 trials are depicted in panel (**A**).

same connection probability $f_w = 0.6$ between neurons as the BTSP networks that we consider. We also used the same sparse input patterns for all models, unless stated otherwise. Note that the sparsity of input patterns tends to increase the capacity of HFNs according to ref. [29], especially for fully connected HFNs[28].

We show capacity results in Fig. S6 for a scaled-up version of the BTSP network, with 100 times longer input patterns, whose length is in the range of the size of area CA3 in the human brain. We simultaneously reduced there the fraction of input bits that have value 1 by the factor 1/100, so that on average the same number of input bits have value 1 as in our standard setting.

Our theory predicts excellent recall capability for this model, even if up to a 2/3 fraction of a memory item is masked in a cue for up to 800,000 memory items. We were not able to carry out numerical simulations for a network of this size.

An important functional benefit of the robust recall that is achieved by memory systems created with BTSP for masked input patterns is demonstrated in Fig. 3F. We considered here the robustness of downstream classifications of memory traces for the case where such downstream classification is defined by a generic thresholded linear readout, with randomly chosen integer readout weights, uniformly sampled from the range -8 to +8, and a fixed threshold 0 of the readout neuron. Robustness was measured in terms of the fraction of memory items $\boldsymbol{x}$ for which masking of the memory item produced an error, i.e., changed the classification result. We show in Fig. 3F that the noise reduction that BTSP achieves according to Fig. 3B in the mapping of memory items to memory traces causes high robustness of such a generic downstream classification. One sees in Fig. 3G that such a downstream classification is substantially less robust when memory traces were created through a RP. Figure 3H shows that the robustness of HFNs with continuous weights is, from this perspective, comparable to that of BTSP with binary weights, with higher robustness for a small number of input patterns but less robustness for a larger number of input patterns.

**Memory traces for overlapping patterns**

Episodic and conjunctive memory requires the storage of memory items that may have common components and are, therefore, far from being orthogonal. Hence, the question arises whether BTSP is able to create functionally useful memory traces also for memory items that have a substantial fraction of common components or features, i.e., common 1's. For independent memory items, the expected number of common 1's is 0.625, i.e., 0.5% of their expected number of 1's. We show in Fig. 4A and B that BTSP can store and separate memory items that have up to 30% of common 1's. In fact, good recall of memory traces can be achieved with incomplete cues, where 33% of the bits are masked. We show in Fig. 4B that the theory for BTSP that we have developed, in particular Eqs.S7 and S8, also works for overlapping memory items, and produces very similar results as the numerical simulations (see Section S3 of the Supplement for details).

According to ref. [30] one can also store a substantial number of overlapping patterns in HFNs, but this requires offline training and continuous weights.

**BTSP endows memory traces with attractor properties that are beyond the reach of RPs**

We had already shown in Fig. 3 that memory traces that are created through BTSP have an inherent attractor property that cannot be achieved by RPs: Recall with partial cues $\boldsymbol{x}'$, where a significant fraction of the 1's in a stored memory item $\boldsymbol{x}$ are masked, activates a memory trace $\boldsymbol{z}(\boldsymbol{x}')$ whose HD to $\boldsymbol{z}(\boldsymbol{x})$ is in general substantially smaller than the HD between $\boldsymbol{x}$ and $\boldsymbol{x}'$. In Fig. S3A, we show that this also holds when the recall cue $\boldsymbol{x}'$ is a randomly perturbed version of $\boldsymbol{x}$, where not only 1's are replaced by 0's but also 0's are replaced by 1's. In contrast, memory traces created through a RP have no such attractor property according to Fig. 3D.

We are addressing here the question whether this attractor quality that is induced by BTSP can also produce input reconstruction, i.e., the characteristic property of a CAM. To test that, one needs to add a generative component to the model, i.e., synaptic connections from the output layer back to the input layer. These feedback connections mimic synaptic connections in the brain that enable the reconstruction of memory content in the neocortex from a memory trace in area CA1. Since the biological implementation of these feedback connections and their synaptic plasticity remain opaque, we have chosen here the simplest possible model: Random feedback connections from the memory neurons to the input neuron layer, see the right parts of panels A and B of Fig. 5, that are subject to a one-shot Hebbian synaptic

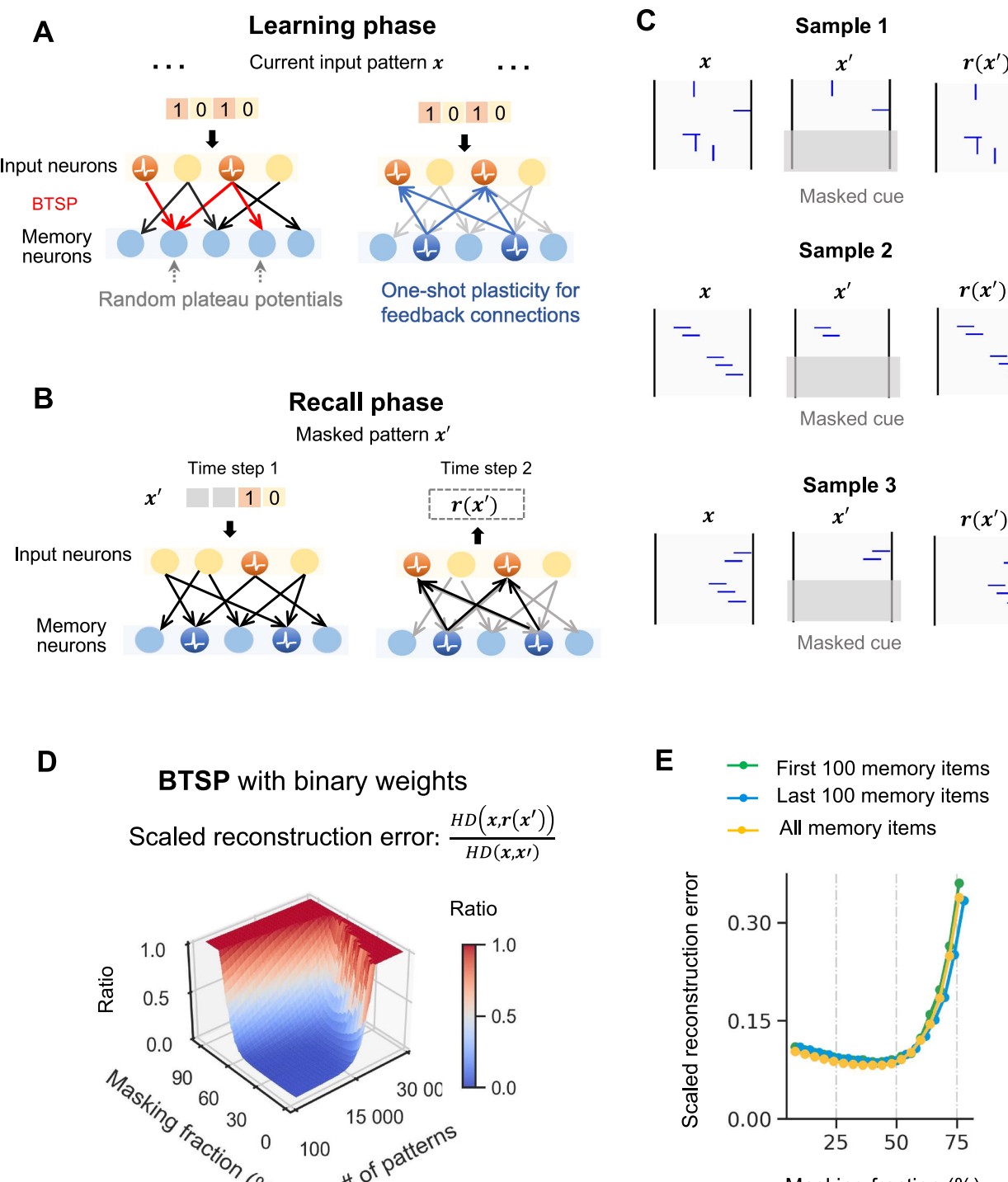

**Fig. 5 | BTSP enables input reconstruction when feedback connections are added. A** Illustration of the model during the learning phase. For each memory item $x$, the forward connections are trained through one-shot learning via BTSP in a single step, as before. Before the next memory item is processed, the feedback connections are trained in a single step through Hebbian-like plasticity (see "*Methods*" for details). **B** Illustration of input reconstruction after learning with BTSP. Input neurons encode a partially masked cue $x'$ at one step and are overwritten by the internally generated pattern $r(x')$ at the next time step. (**C**) Demonstration of input completion via BTSP for an ensemble of 2000 input patterns that are easy to visualize: The patterns were collections of 5 line segments (each 8 pixels long) in horizontal or vertical direction, in random positions. A generic pattern of this type is shown as the first plot. During the recall phase, the last 20 rows of each original image were masked (indicated by the grey area). To make input completion more challenging, we included in the list of 2000 memory items also a pair of patterns that agreed in 3 of the 5 line segments, as shown in the 2nd and 3rd plot. BTSP with feedback connections effectively distinguishes and accurately reconstructs masked versions of these two very similar memory items. **D** shows a quantitative analysis of the input reconstruction capability of the BTSP model for the same ensemble of random input patterns as used in Fig. 3. We used the scaled reconstruction error $HD(r(x'), x)/HD(x, x')$ for memory items $x$ that occurred during the learning phase and the internally generated pattern $r(x')$ for a masked version $x'$ of this memory item as a cue. **E** The scaled reconstruction error is for memory traces created through BTSP about the same for the first 100 and the last 100 memory items in a list of 10.000 items, measured for both after all 10,000 memory items have been learnt. Average results over 5 trials are depicted in panels D and E.

plasticity rule during the learning phase (see "*Methods*"). The neurons in the input layer are modelled like the neurons in the output layer as McCulloch-Pitts neurons. The resulting input reconstruction capability after one-shot learning with BTSP and the Hebbian plasticity is illustrated in Fig. 5C. Input completion for a cue $x'$ is produced immediately at the next time step after the presentation of a cue $x'$: The cue $x'$ is then removed, and the input neurons are driven by feedback from the output layer. An input neuron now assumes the value 1 if its weighted sum of inputs from the output layer exceeds its firing threshold. We write $r(x')$ for the bit vector that results in this way from the input cue $x'$ at the preceding time step. If the recall cue $x'$ is a masked version of a memory item $x$ that occurred during learning, the internally generated pattern $r(x')$ approximates in general the original memory item $x$ very well, see the first panel of Fig. 5C.

We analyze this input reconstruction capability quantitatively in Fig. 5D and Fig. S7, for the same sparse input patterns as in Fig. 3. We see in Fig. 5D that the $HD(x, r(x'))$ between an original memory item $x$ and the internally generated completion $r(x')$ of a masked version $x'$ of the memory item is in general substantially smaller than the HD between $x$ and $x'$, even when large numbers of memory items $x$ are stored, and if a large fraction of its bits is masked in the recall cue $x'$.

A comparison between Fig. 5D and E indicates that input reconstruction is for the CAM with binary weights that is created through BTSP about as good as the one offered by HFNs with continuous weights, for the sparsely represented memory items and incomplete network connectivity that are considered throughout this paper. Furthermore, we show in section S12 of the Supplement that HFNs with binary weights provide no viable CAM alternative for the same set of input patterns as predicted by theoretical results[11].

Figure 5 E shows that the scaled reconstruction error is almost the same for the first 100 and the last 100 in a list of 10,000 memory items that are subsequently learnt by BTSP. This is insofar remarkable as the previously proposed rule for one-shot learning[31] (we are referring to their model SP) exhibits a substantial degradation of memory traces for earlier learnt memory items, according to Fig. 4 a, b in their paper. The source for this marked difference is that the plasticity rule[31] depends on postsynaptic firing (instead of stochastic gating signals), which induces recruitment of the same neurons for many memory traces.

These positive results for BTSP provide good news for the improvement of CAMs in neuromorphic hardware. CAM models based on HFNs have already been implemented through crossbar arrays of memristors in neuromorphic hardware[17–19], using off-chip learning of non-binary synaptic weights. A comparison of Fig. S7A and C suggests that for sparse input patterns and sparse network connectivity, at least the same CAM quality as HFNs with continuous weights, can be achieved by crossbar arrays of memristors that have just two resistance states, using on-chip learning via BTSP. In addition, BTSP enables instant memory recall, whereas a HFN requires multiple iterations before the network state converges to an attractor and, therefore, takes more time for input completion. We show in Fig. S8 that the input completion performance with BTSP becomes even better for a lower value of $f_q$.

### BTSP induces a repulsion effect for closely related memory items

Curiously, BTSP induces an effect that is seemingly incompatible with the previously described attractor properties for similar memory items: Memory traces for similar memory items are pulled away from each other. A closer look shows that this repulsion effect is, in fact, compatible with the previously demonstrated attractor properties of BTSP-induced memory traces: The difference is that these similar memory items occur during learning, not only during subsequent recall. Specifically, we show that the LTD part of BTSP makes the

memory trace dissimilar to that of a previously processed similar memory item. So far, the LTD part of BTSP was always running in the background and gave rise to the desirable effect that the number of strong synaptic connections remained limited (see Fig. 2D). Here, we elucidate another important functional impact of the LTD part of BTSP.

An obvious functional advantage of this repulsion effect is that it enables differential downstream processing of memory items that differ in just a few features. Arguably, this is a key trait of higher cognitive function. In fact, experimental studies of memory traces in the human brain, especially in area CA1, show that the human brain exhibits a repulsion effect[12–14,32]. However, previous memory models have not been able to reproduce this. In particular, a RP produces an opposite effect, whereby memory traces for more similar memory items have more similar memory traces, see Fig. 6A. To the best of our knowledge, BTSP provides the first learning-based model for the repulsion effect of human memory.

Figure 6 B shows results of numerical simulations for an experiment with BTSP that mimics the experimental design of studies of the repulsion effect in the human brain[12–14,32]: A sequence of $M$ memory items is learned through BTSP, where these memory items are pairwise unrelated except for two that have a large overlap ratio of 40%, but may appear at any position in the sequence. We evaluate the similarity of resulting memory traces similarly as in these experiments, where the Pearson correlation coefficient was applied to voxels of fMRI recordings: We measure the overlap ratio of two memory traces by their fraction of common 1's, divided by the average number of 1's in a memory trace. Also the similarity (overlap) of two memory items is measured by using the fraction of common 1's. Whereas no repulsion effect occurs with RPs according to Fig. 6A, a significant repulsion effect is induced according to Fig. 6B by BTSP. Note that its strength is in this model comparable to that in the human brain, see Fig. 3C[12].

One can, in fact, clearly understand how the LTD part of BTSP induces this repulsion effect. If a memory neuron receives during learning plateau potentials for both of two similar memory items, the strength of most of its synaptic connections from neurons that are active in both memory items will have a value 0 after learning: These synaptic weights are first increased and then decreased by the LTD part of BTSP. This effect reduces the overlap of the resulting memory traces because this memory neuron will not belong to the memory trace for either of the two similar memory items. On the other hand, repeated plateau potentials to the same memory neuron for two memory items that have largely non-overlapping sets of 1's will only reduce weights from those few input neurons that assume value 1 for both input patterns. Therefore, this memory neuron is more likely to fire for both memory items after learning has been completed, thereby increasing the overlap ratio of the resulting memory traces (see the right part of Fig. 6B).

## Discussion

We have created and examined a simple rule for BTSP, the recently discovered one-shot learning rule in the brain[1–3], see Fig. 1D and E. Whereas previous models for BTSP aimed at reproducing experimental data from mice during navigation in mazes[7,8], we have focused here on functional implications of the simple BTSP rule for the creation of memory systems for large numbers of generic sparsely encoded memory items. More detailed BTSP data and rules with continuous weights can be reproduced by this simple rule when multiple synaptic connections are taken into account, see Fig. 1F and G. Importantly, this BTSP rule is sufficiently simple so that its functional impact can be studied analytically.

One of the most surprising and intriguing features of BTSP is that it does not rely on postsynaptic neural activity, like STDP, Hebbian rules, and virtually all other synaptic plasticity rules that have been studied in the context of memory models or elsewhere in theoretical neuroscience. Instead, it relies on external stochastic gating signals

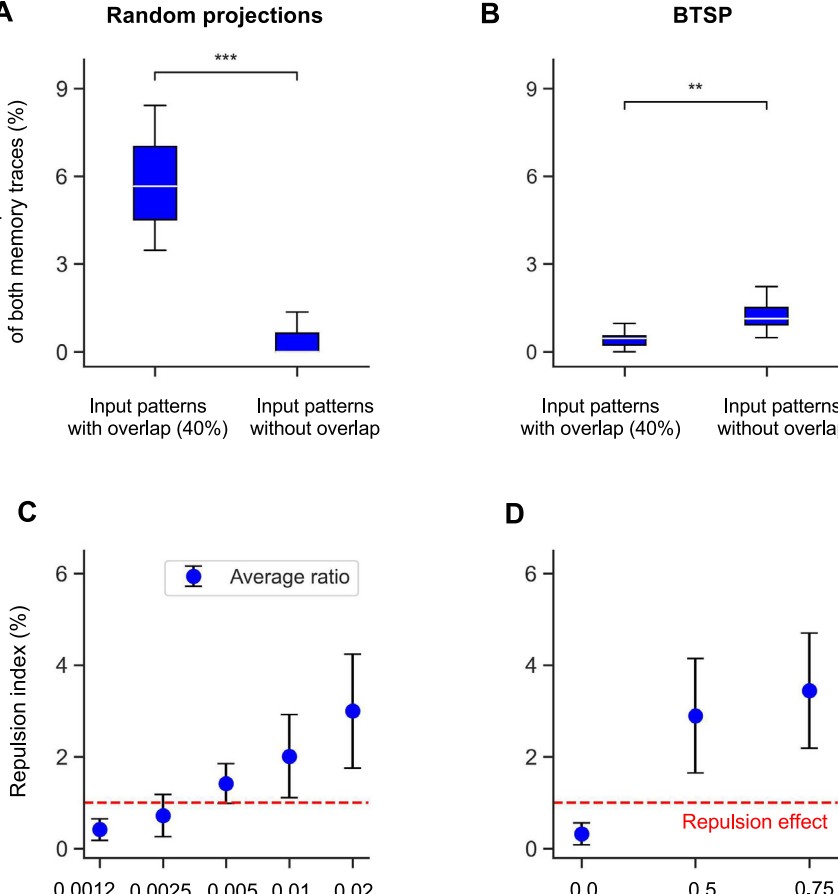

**Fig. 6 | In contrast to random projections, the BTSP rule can reproduce the repulsion effect of human memory. A** The overlap ratio of the memory traces created by a RP for a pair of memory items with 40% overlap is shown on the left, and for a pair of unrelated memory items on the right. The overlap ratio is higher for the more similar memory items (***$p < 0.001$, two-tailed one-sample t-test). **B** The same analysis was carried out for memory traces that resulted from BTSP. Here, the overlap ratio is smaller for more similar memory items (**$p < 0.01$, two-tailed one-sample t-test), like in corresponding experimental data from the human brain[12]. Details of the data generation are provided in the *Methods*; $f_q = 0.02$ was used in panel (**B**). **C** and **D** elucidate the impact of the plateau potentials probability $f_q$ and of the LTD probability (default value 0.5) on the repulsion effect. The

strength of repulsion was measured by the repulsion index, which is obtained by dividing the overlap ratio of memory traces for a pair of unrelated memory items by the overlap ratio for a pair of memory items with 40% overlap. Values above the red dashed lines that are defined by value 1 indicate the occurrence of the repulsion effect. One sees that values for $f_q$ that are below our biologically derived default value 0.005 do not produce repulsion and also that the LTD component of BTSP is essential for repulsion. In fact, the repulsion effect becomes even stronger when the LTD probability is increased beyond its default value. The error bars in both panels represent the standard deviation of the overlap ratio over 20 trials with new generations of random connection weight, input patterns, and plateau potentials.

that open the gate for synaptic plasticity in a neuron for several seconds. A generic functional advantage of this mechanism is that it spreads the "memory load" uniformly over all neurons in the network, thereby avoiding degradation of previously generated memory traces, see Fig. 5E.

Because of the reliance of BTSP on stochastic gating signals, the probability of the occurrence of these gating signals, i.e., the parameter $f_q$, is an essential parameter for BTSP that has no analogue in previously considered learning rules. We have analyzed the functional impact of $f_q$, both theoretically and through numerical simulations, see Figs. 2C, D, E, and 6 C, and Fig. S8. Our results suggest that the experimentally determined value $f_q = 0.005$ can be seen as the sweet spot for combining several functionally attractive features of BTSP.

We have shown that the resulting predictions agree very well with the results of numerical simulations. It also reproduces the primary functional impact of BTSP that had been reported in ref. 1: Instantaneous creation of place fields in a CA1 neuron as a result of a

plateau potential (Fig. 1D). Furthermore, our BTSP rule reproduces an effect that was highlighted in the study[1] as the hallmark of its plasticity window on the time scale of seconds, rather than the time scale of milliseconds that is characteristic for STDP: A substantial increase in the spatial extension of the place field in the case of higher running speed during plasticity induction, see Fig. 1E. In other words, the CA1 neuron is induced to fire not only for the current spatial location at the onset of a plateau potential, but also for locations that were traversed several seconds before and after that onset. If one views the spatial locations that are traversed during this seconds-long plasticity window as frames of an episodic memory, one sees that BTSP can create memory traces for episodes that are several seconds long.

Another functional impact of the seconds-long plasticity window of BTSP is that it makes the traces for such memory traces in area CA1 sufficiently large: It enables synaptic plasticity during an episode in sufficiently many CA1 neurons (see Fig. 2D for a theoretical analysis), in spite of the experimentally found very low rate of plateau potentials

0.0005 per CA1 neuron per second[6]. In other words, it brings the probability $f_q$ that a presynaptic input occurs within the plasticity window that is opened by a plateau potential into a range that is especially suited for creating functionally useful memory traces.

Furthermore, online learning with BTSP also creates well-working CAM with binary weights if the memory items are overlapping. Hence, this CAM can also be used to store episodic memories or conjunctive memories that have common components. Finally, we have shown in Fig. 6 that BTSP reproduces the repulsion effect of the human brain that pulls memory traces for similar memory items apart, as documented in refs. 12–14,32. This effect can be seen as a vital component of higher cognitive function since it enables differential processing of experiences that differ only in a few but possibly salient features. However, the repulsion effect could not be reproduced with previously considered learning rules.

On a more abstract level, our results show that the stochastic gating of BTSP can be seen as a method for porting some functionally attractive properties of the fly algorithm[15] into a learning system. More precisely, BTSP embeds these advantages of stochasticity into a learning mechanism that tailors the resulting "random projection" to the particular ensemble of memory items that are to be stored. We have shown that this makes BTSP superior to the fly algorithm and random projections as methods for neural coding of input patterns because BTSP induces powerful attractor properties for the memory traces that it creates, see Figs. 3, 4, 5.

Another biological mechanism for one-shot creation of memory traces in a biological organism had been described in Imam and Cleland (2020)[33] for the mammalian olfactory bulb. This work has in common with our approach that it also provides one-shot insertion of memory items into a neural network that supports recall with noisy cues. Furthermore, they consider a spiking neural network implementation. Spike-based implementations of HFNs, where the timing of spikes conveys analog information, had already been introduced by ref. 34. In principle, this model[34] also allowed one-shot learning, because the standard learning rule for HFNs is a one-shot learning rule. But both the HFN model and Imam and Cleland (2020) require that synaptic weights can assume a large number of values, whereas our BTSP-based model requires only binary weight values. Another difference to Imam and Cleland's model[33] is that in their approach new neurons have to be added to the networks when new memory items are to be learnt. In contrast, our memory network can have a fixed architecture. Additionally, in terms of learning rules, two different mechanisms are employed in their model: A simplified STDP rule for synaptic weights and a less standard rule for modifying the blocking period for firing that is caused by a spike of the inhibitory GC neuron in the postsynaptic MC cell. Since they add new GC neurons whenever a new memory item is to be learnt, their model is likely to avoid the degradation of earlier memory traces that typically occurs with Hebbian-like plasticity rules for one-shot learning. In contrast, the stochastic nature of gating signals for BTSP, in conjunction with sparse coding, reduces the chance that previously created memory traces are overwritten, see our explanations for Fig. 5E.

Our BTSP rule is, to the best of our knowledge, the first online learning rule that is able to create a functionally powerful CAM with binary synaptic weights. Furthermore, updates of the CAM can be carried out through a simple one-shot learning rule that is within reach for on-chip learning. Hence, the creation of memory systems through BTSP provides an attractive alternative to HFNs for modelling in computational neuroscience and for creating CAMs in neuromorphic hardware. HFNs require weights whose number of distinguishable values grows linearly with the number of memory items that are stored. Results of numerical simulations in Fig. S9 confirm theoretical predictions from ref. 11 that the memory capacity of HFNs drops drastically if weights are constrained to binary values. But both the number of different values that a synaptic weight in a biological neural network[35], and the number of distinguishable resistance states that a memristor can assume are severely limited. Furthermore, memristors whose resistance just has to assume a small number of values are substantially easier to fabricate and take less space on the chip. Therefore, CAMs that require just binary synapses are especially attractive from this perspective. Previous neuromorphic implementations of CAM were based on the HFN paradigm, and could therefore store only a very small number of memories[17–19]. Hence, CAMs based on BTSP are likely to substantially advance the state of the art for neuromorphic CAM.

A substantially more ambitious goal would be to create CAMs with superlinear, or even exponential capacity in neuromorphic hardware. Modern Hopfield networks and dense associative memories[36,37] as well as bipartite Hopfield networks[38] provide interesting theoretical models for that. However, a closer look shows that the latter is not able to store externally given memory items, and that the former requires, like the classical HFN, synaptic weights that can assume a very large number of distinguishable values. Furthermore, the memory capacity of modern Hopfield networks and dense associative memory networks remains linear in the network size according to ref. 37, unless one moves to networks with higher order interactions among neurons, i.e., if one goes beyond pairwise interaction of neurons via synapses. Obviously, this will require a completely new technology in order to create an energy-efficient neuromorphic implementation of such a network.

This work suggests an alternative research direction, where new insight from the brain paves the road for the design of substantially more energy-efficient content-addressable memory systems. These require just binary weights, and can be updated through one-shot on-chip synaptic plasticity.

## Methods

### A learning theory for BTSP

**Background and preparations for memory trace estimates.** For simplicity, our theoretical analysis will be presented using the BTSP rule from Eq. (1), under the assumption that the weight update is made independently for each synapse with a probability of 0.5. This methodological assumption aims to simplify the analytical framework and improve the comprehensibility of the results without compromising the integrity of our findings. As substantiated by the numerical simulations and theoretical predictions presented in Figs. S1B and C, this approach demonstrates consistency with the broader results.

We will employ the properties of the binomial distribution for the memory trace estimates. Key parameters used in the following derivations are summarized in Table 1. Additionally, we introduce the following lemma crucial for handling the parity of weight updates in Eqs. (12), (13), (22) and (23).

**Lemma 1.** Let $f(k_x, N_x, p_x)$ be the probability of binomial distribution that a random variable $X$ achieves $k_x$ success in $N_x$ Bernoulli trials. Then, the probability that $X$ is even can be given by

$$P(X \text{ is even}) = \frac{1}{2}\left(1 + (1 - 2p_x)^{N_x}\right). \tag{2}$$

Similarly, the probability that $X$ is odd can be given by

$$P(X \text{ is odd}) = \frac{1}{2}\left(1 - (1 - 2p_x)^{N_x}\right). \tag{3}$$

**Table 1 | Definitions and explanations of key parameters used in the theoretical analysis**

| Variables | Category | Explanations | Definitions |
|---|---|---|---|
| $f(*)$ | General notations | Probability mass function of the binomial distribution | |
| $F(*)$ | | Cumulative distribution function of the binomial distribution | |
| $w_{ji}$ | | The synaptic weight between memory neuron $j$ and input neuron $i$, which can be in an active state (1) or inactive state (0). | |
| $Q(\boldsymbol{x})$ | | Set of indices for memory neurons that receive a plateau potential for a given pattern $\boldsymbol{x}$ during learning | |
| $supp(\boldsymbol{x})$ | | Set which collects positions of 1's in $\boldsymbol{x}$ | |
| $l(\boldsymbol{x})$ | | Size of $supp(\boldsymbol{x})$ | |
| $S(j)$ | | Set which collects the index of input patterns (presented sequentially) for which the memory neuron $j$ receives plateau potentials during learning | |
| $\boldsymbol{y}(j)$ | | The unit vector with only $j$-th bit position being 1 | |
| $c$ | | Number of common 1's among overlapping memory items | |
| $f_q$ | | Probability of plateau potential | |
| $f_w$ | | Probability of connections | |
| $f_d$ | | Fraction of 1's being masked to zeros | |
| $f_p$ | Random pattern estimation | Input density | |
| $p_e$(or $p_o$) | | Conditional probability that a synapse connecting 1's of $\boldsymbol{x}$ and a memory neuron remains in an active state during the recall phase, given that the memory neuron receives (or does not receive) a plateau potential for $\boldsymbol{x}$ | Eqs. (12) and (13) |
| $p_{f1}$(or $p_{f2}$) | | Conditional probability that a memory neuron fires in the recall phase when exposed to $\boldsymbol{x}$, given that the memory neuron receives (or does not receive) plateau for $\boldsymbol{x}$ during learning | Eqs. (14) and (15) |
| $p_{f_{avg}}$ | | Probability that a memory neuron fires in the recall phase when exposed to $\boldsymbol{x}$ | Eq. (16) |
| $\hat{f}_p$ | Overlapping pattern estimation | Input density of the remaining bits in overlapping memory items, excluding the common 1's | |
| $\hat{p}_e$( or $\hat{p}_o$) | | Conditional probability that a synapse connecting 1's of $\boldsymbol{x}$ and a memory neuron keeps in an active state in the recall phase, given that the memory neuron receives (or does not receive) plateau for $\boldsymbol{x}$ during learning | Eqs. (22) and (23) |
| $\hat{p}_{f1}$( or $\hat{p}_{f2}$) | | Conditional probability of a memory neuron firing in response to $\boldsymbol{x}$ in the recall phase, given that the memory neuron receives (or does not receive) plateau for $\boldsymbol{x}$ during learning | Eqs.S2 and S3 |
| $\hat{p}_{f_{avg}}$ | | Probability that a memory neuron fires in the recall phase when exposed to $\boldsymbol{x}$ | Eq.S4 |

**Proof**. Note that

$$((1-p_x)+p_x)^{N_x} = \sum_{k_b=0}^{N_x} C_{N_x}^{k_b} p_x^{k_b} (1-p_x)^{N_x-k_b} \tag{4}$$
$$= P(X=0) + P(X=1) + \cdots + P(X=N_x),$$

$$((1-p_x)-p_x)^{N_x} = \sum_{k_b=0}^{N_x} C_{N_x}^{k_b} (-p_x)^{k_b} (1-p_x)^{N_x-k_b} \tag{5}$$
$$= P(X=0) - P(X=1) + \cdots + (-1)^{N_x} P(X=N_x).$$

Combining the two equations then gives

$$P(X \text{ is even}) = \frac{1}{2}\left(1 + (1-2p_x)^{N_x}\right), \tag{6}$$

$$P(X \text{ is odd}) = \frac{1}{2}\left(1 - (1-2p_x)^{N_x}\right). \tag{7}$$

**Estimates of properties of memory traces for randomly drawn memory items**
**Estimates of firing probability of memory neurons after learning M randomly drawn memory items.** To estimate the HD of memory traces for different input patterns, we need to estimate the probability that a memory neuron fires differently for different input patterns. To obtain this, we start with deriving a basic estimate called the firing

probability, which represents the probability that a memory neuron fires in response to the memory item $\boldsymbol{x}$ after learning $M$ patterns. Note that the firing behaviour of a memory neuron is determined by whether the weighted sum of binary inputs is greater than a firing threshold $v_{th}$. It indicates that the firing probability depends on the number of weights that are in the active status (i.e., $w_{ji}=1$) after the learning is completed. Therefore, to estimate this firing probability, we need to model the effect of BTSP on the active state of $w_{ji}$ after learning.

We introduce the unit vector $\boldsymbol{y}(j) \in \mathbb{R}^{n \times 1}$ with only the $j$ bit position as 1 for representing the binary plateau pattern. For simplicity, we assume that the initial weights are set to a zero matrix and use the term $\boldsymbol{W}$ to refer to the weights after learning $M$ patterns. The weight matrix $\boldsymbol{W}$ can be written as the summation of multiple weight updates, organized in the order of the bit position $j$, as follows:

$$\boldsymbol{W} = \sum_{j=1}^{n} \boldsymbol{y}(j) \times \hat{1}\left(\sum_{k \in S(j)} \boldsymbol{x}_k^T\right), \tag{8}$$

where the set $S(j)$ collects the index of patterns for which the memory neuron $j$ receives the plateau potentials during the learning process. $\hat{1}(x)$ is an indicator function which models the impact of the parity of weight update:

$$\hat{1}(x) = \begin{cases} 1, & \text{if } x \text{ is odd}, \\ 0, & \text{if } x \text{ is even}. \end{cases} \tag{9}$$

Next, we calculate the weighted sum of inputs $\boldsymbol{u}(\boldsymbol{x})$:

$$\boldsymbol{u}(\boldsymbol{x}) = \boldsymbol{W} \times \boldsymbol{x} = \sum_{j=1}^{n}\left(\boldsymbol{y}(j)\times\hat{1}\left(\sum_{k\in S(j)}\boldsymbol{x}_k^T\right)\right)\times\boldsymbol{x} = \sum_{j=1}^{n}\boldsymbol{y}(j)\times\left(\hat{1}\left(\sum_{k\in S(j)}\boldsymbol{x}_k^T\right)\times\boldsymbol{x}\right). \tag{10}$$

where $\boldsymbol{x}_k$ represents the memory item for which the memory neuron $j$ receives a plateau potential during the learning phase.

Let $u_j$ be the $j_{th}$ element of $\boldsymbol{u}$. According to the definition of $\boldsymbol{y}(j)$, we have

$$u_j(\boldsymbol{x}) = \hat{1}\left(\sum_{k\in S(j)}\boldsymbol{x}_k^T\right)\times\boldsymbol{x} = \sum_{i\in supp(\boldsymbol{x})}\hat{1}\left(\sum_{k\in S(j)}x_{ki}\right), \tag{11}$$

where $x_{ki}$ is the element $i$ in $\boldsymbol{x}_k$.

Note that $\hat{1}(\sum_{k\in S(j)}x_{ki})$ is a binary variable that represents the active state of $w_{ji}$. According to Eq. (1), its value is determined by the parity of actual weight updates during the learning phase. Due to the assumption of the input density $f_p$, plateau probability $f_q$, and the weight update probability (0.5), the probability of weight update can be modeled by Bernoulli distribution with a probability of $f_p f_q/2$. Therefore, we can estimate the probability $P(\hat{1}(\sum_{k\in S(j)}x_{ki})=1)$ in the following two cases:

- For any $j_1 \in Q(\boldsymbol{x})$ and any $i \in supp(\boldsymbol{x})$, we estimate the probability $p_e$ that $w_{j_1 i}$ is active after learning $M$ patterns:

$$p_e := P\left(\hat{1}\left(\sum_{k\in S(j_1)}x_{ki}\right)=1|\boldsymbol{x}\right) = P\left(\sum_{k\in S(j_1)}x_{ki}\text{ is odd }|\boldsymbol{x}\right)$$
$$= P\left(\sum_{k\in S(j_1)}x_{ki}-1\text{ is even}\right) = \frac{1}{2}\left(1+(1-f_p f_q)^{M-1}\right), \tag{12}$$

where the estimate of the parity in the last equality uses Lemma 1.

- For any $j_2 \notin Q(\boldsymbol{x})$ and any $i \in supp(\boldsymbol{x})$, we estimate the probability $p_o$ that $w_{j_2 i}$ is active after learning $M$ patterns:

$$p_o := P\left(\hat{1}\left(\sum_{k\in S(j_2)}x_{ki}\right)=1|\boldsymbol{x}\right) = P\left(\sum_{k\in S(j_2)}x_{ki}\text{ is odd }|\boldsymbol{x}\right)$$
$$= \frac{1}{2}\left(1-(1-f_p f_q)^{M-1}\right). \tag{13}$$

The above equations provide estimates of the probability that any given connection is in an active state after learning $M$ patterns. In order to incorporate the effects of the inherited connection probability $f_w$, we directly multiply $f_w$ by $p_o$ (or $p_e$) since they are independent. We then employ the cumulative binomial probability $F$ to estimate the firing probability of memory neurons given the size of support set $\iota(\boldsymbol{x})$ as $l$:

- For any $j_1 \in Q(\boldsymbol{x})$, we have:

$$p_{f1}(j_1,\boldsymbol{x},l) := P\left(u_{j_1}>v_{th}|\boldsymbol{x},l,j_1\in Q(\boldsymbol{x})\right) = 1-F(v_{th},l,f_w p_e). \tag{14}$$

- For any $j_2 \notin Q(\boldsymbol{x})$, we have:

$$p_{f2}(j_2,\boldsymbol{x},l) := P\left(u_{j_2}>v_{th}|\boldsymbol{x},l,j_2\notin Q(\boldsymbol{x})\right) = 1-F(v_{th},l,f_w p_o). \tag{15}$$

The above Eqs. (14) and (15) show that the formulas of firing probability do not depend on the specific positions of $j_1$ and $j_2$. For brevity, we will omit the indices $j_1$ and $j_2$ of the brackets in the following derivations.

Given the estimates of $p_{f1}$ and $p_{f2}$, we can finally obtain the firing probability of a memory neuron by applying the total probability rule, which allows us to relax the condition of the given size $\iota(\boldsymbol{x})$ of the support set:

$$p_{f_{avg}}(\boldsymbol{x}) = P(u_j>v_{th}|\boldsymbol{x})$$
$$= \sum_{l=0}^{m}P(u_j>v_{th},\iota(\boldsymbol{x})=l|\boldsymbol{x})$$
$$= \sum_{l=0}^{m}P(\iota(\boldsymbol{x})=l)P(u_j>v_{th}|\iota(\boldsymbol{x})=l,\boldsymbol{x})$$
$$= \sum_{l=0}^{m}P(\iota(\boldsymbol{x})=l)\Big(P(j\in Q(\boldsymbol{x}))P(u_j>v_{th}|j\in Q(\boldsymbol{x}))$$
$$+P(j\notin Q(\boldsymbol{x}))P(u_j>v_{th}|j\notin Q(\boldsymbol{x}))\Big)$$
$$= \sum_{l=0}^{m}C_m^l f_p^l(1-f_p)^{m-l}\Big(f_q p_{f1}(\boldsymbol{x},l)+(1-f_q)p_{f2}(\boldsymbol{x},l)\Big). \tag{16}$$

**Estimates of expected HD between memory traces for the original and masked memory items.** In this section, we utilize the properties of $p_e$ and $p_o$ to estimate the expected HD between memory traces for the original and masked memory items.

Suppose $\boldsymbol{x}$ is a random binary memory item with the input density of $f_d$ and $\boldsymbol{x}'$ is the masked pattern of $\boldsymbol{x}$ with the fraction $f_d$ of 1's being masked to zeros. We formulate the expected HD in terms of the probability that individual neurons fire differently for any complete memory item and its partially masked pattern:

$$E(HD(\boldsymbol{z}(\boldsymbol{x}),\boldsymbol{z}(\boldsymbol{x}'))) = \sum_{j=1}^{n}P(z_j(\boldsymbol{x})\neq z_j(\boldsymbol{x}'))$$
$$= \sum_{j=1}^{n}P(j\in Q(\boldsymbol{x}))P\Big(z_j(\boldsymbol{x})\neq z_j(\boldsymbol{x}')|j\in Q(\boldsymbol{x})\Big)$$
$$+P(j\notin Q(\boldsymbol{x}))P\Big(z_j(\boldsymbol{x})\neq z_j(\boldsymbol{x}')|j\notin Q(\boldsymbol{x})\Big)$$
$$= nf_q P\Big(z_j(\boldsymbol{x})\neq z_j(\boldsymbol{x}')|j\in Q(\boldsymbol{x})\Big)$$
$$+n(1-f_q)P\Big(z_j(\boldsymbol{x})\neq z_j(\boldsymbol{x}')|j\notin Q(\boldsymbol{x})\Big) \tag{17}$$

where $Q(\boldsymbol{x})$ refers to the set of indices of the memory neuron that got plateau potentials for $\boldsymbol{x}$ during the learning process. Eq. (17) indicates that to estimate this expression $E(HD(\boldsymbol{z}(\boldsymbol{x}),\boldsymbol{z}(\boldsymbol{x}')))$, we only need to estimate the probability that a memory neuron fires differently for $\boldsymbol{x}$ and $\boldsymbol{x}'$ for both the set $Q(\boldsymbol{x})$ and its complement.

By using $p_{f_1}$ and $p_{f_2}$ as defined by Eqs. (14) and (15), we estimate $P(z_j(\boldsymbol{x})\neq z_j(\boldsymbol{x}'))$ in the following two separate cases:

- For any $j_1 \in Q(\boldsymbol{x})$, we have:

$$P\Big(z_{j_1}(\boldsymbol{x})\neq z_{j_1}(\boldsymbol{x}')|j_1\in Q(\boldsymbol{x}),\iota(\boldsymbol{x})=l\Big)$$
$$= P\Big(z_{j_1}(\boldsymbol{x})=1,z_{j_1}(\boldsymbol{x}')=0|j_1\in Q(\boldsymbol{x}),\iota(\boldsymbol{x})=l\Big)$$
$$+P\Big(z_{j_1}(\boldsymbol{x})=0,z_{j_1}(\boldsymbol{x}')=1|j_1\in Q(\boldsymbol{x}),\iota(\boldsymbol{x})=l\Big)$$
$$= P\Big(z_{j_1}(\boldsymbol{x})=1,z_{j_1}(\boldsymbol{x}')=0|j_1\in Q(\boldsymbol{x}),\iota(\boldsymbol{x})=l\Big)$$
$$\text{(Note that }P(z_{j_1}(\boldsymbol{x})=0,z_{j_1}(\boldsymbol{x}')=1)=0)$$
$$\approx P\Big(z_{j_1}(\boldsymbol{x})=1|j_1\in Q(\boldsymbol{x}),\iota(\boldsymbol{x})=l\Big)P\Big(z_{j_1}(\boldsymbol{x}')=0|j_1\in Q(\boldsymbol{x}),\iota(\boldsymbol{x})=l\Big)$$
$$= (1-F(v_{th},l,f_w p_e))F(v_{th},l(1-f_d),f_w p_e). \tag{18}$$

The above estimate approximates $P(z_{j_1}(\boldsymbol{x}) = 1, z_{j_1}(\boldsymbol{x}') = 0)$ as two independent terms, $P(z_{j_1}(\boldsymbol{x}) = 1)$ and $P(z_{j_1}(\boldsymbol{x}') = 1)$. It ignores the dependence on the same parity of the weight updates indicated by Eqs. (14) and (15). Nevertheless, our numerical estimates show good consistency using the simple strategy. A more precise but complicated estimate is provided in the derivation for the overlapping memory items (Eqs.S5 and S6), where we further employ the total probability rule to deal with this dependency and make this estimate applicable for the common 1's.

- For any $j_2 \notin Q(\boldsymbol{x})$, similarly, we have

$$P\left(z_{j_2}(\boldsymbol{x}) \neq z_{j_2}(\boldsymbol{x}') | j_2 \notin Q(\boldsymbol{x}), \iota(\boldsymbol{x}) = I\right)$$
$$= P\left(z_{j_2}(\boldsymbol{x}) = 1, z_{j_2}(\boldsymbol{x}') = 0 | j_2 \notin Q(\boldsymbol{x}), \iota(\boldsymbol{x}) = I\right) \quad (19)$$
$$\approx F(v_{th}, I(1 - f_d), f_w p_o)(1 - F(v_{th}, I, f_w p_o)).$$

By combining Eqs. (18) and (19), we can obtain estimates for the expected HD,

$$E(HD(\boldsymbol{z}(\boldsymbol{x}), \boldsymbol{z}(\boldsymbol{x}')))$$
$$= \sum_{I=0}^{m} Pr(\iota(\boldsymbol{x}) = I) E(HD(\boldsymbol{z}(\boldsymbol{x}), \boldsymbol{z}(\boldsymbol{x}')) | \iota(\boldsymbol{x}) = I)$$
$$= \sum_{I=0}^{m} n Pr(\iota(\boldsymbol{x}) = I)\left(f_q P\left(z_{j_1}(\boldsymbol{x}) \neq z_{j_1}(\boldsymbol{x}') | j_1 \in Q(\boldsymbol{x}), \iota(\boldsymbol{x}) = I\right)\right.$$
$$+ (1 - f_q) P\left((z_{j_2}(\boldsymbol{x})) \neq z_{j_2}(\boldsymbol{x}') | j_2 \notin Q(\boldsymbol{x}), \iota(\boldsymbol{x}) = I\right)\bigg)$$
$$\approx n \sum_{I=0}^{m} C_m^I f_p^I (1 - f_p)^{m-I}\left(f_q F(v_{th}, I(1 - f_d), f_w p_e)(1 - F(v_{th}, I, f_w p_e))\right.$$
$$+ (1 - f_q) F(v_{th}, I(1 - f_d), f_w p_o)(1 - F(v_{th}, I, f_w p_o))\bigg).$$
$$(20)$$

$E(HD(\boldsymbol{z}(\boldsymbol{x}), \boldsymbol{z}(\boldsymbol{x}')))$ provides an estimate of dissimilarity between the memory traces for the original and masked memory items, which was used in Fig. 2C.

**Estimates of expected HD between memory traces for different randomly drawn memory items.** To estimate the average HD between memory traces of two different memory items, we use $\boldsymbol{x}_a$ and $\boldsymbol{x}_b$ to represent two different randomly drawn patterns. Owing to the assumption of randomly drawn memory items, we can directly utilize $p_{f_{avg}}$ to estimate $HD(\boldsymbol{z}(\boldsymbol{x}_a), \boldsymbol{z}(\boldsymbol{x}_b))$ as follows,

$$E\left(HD(\boldsymbol{z}(\boldsymbol{x}_a), \boldsymbol{z}(\boldsymbol{x}_b))\right)$$
$$= \sum_{j=1}^{n} P\left(z_j(\boldsymbol{x}_a) \neq z_j(\boldsymbol{x}_b)\right)$$
$$= \sum_{j=1}^{n} P\left(z_j(\boldsymbol{x}_a) = 1, z_j(\boldsymbol{x}_b) = 0\right) + P\left(z_j(\boldsymbol{x}_a) = 0, z_j(\boldsymbol{x}_b) = 1\right) \quad (21)$$
$$= \sum_{j=1}^{n} P(z_j(\boldsymbol{x}_a) = 1) P(z_j(\boldsymbol{x}_b) = 0) + P(z_j(\boldsymbol{x}_a) = 0) P(z_j(\boldsymbol{x}_b) = 1)$$
$$= 2n p_{f_{avg}}(1 - p_{f_{avg}}).$$

Eq. (21) provides the estimates of the memory traces for different original memory items. By combining the estimates in Eq. (20) and Eq. (21), we can estimate the ratio of HD used in Fig. 2C.

**Estimates of properties of memory traces for overlapping memory items.** For overlapping memory items, the approach to estimating firing probabilities is similar to that for randomly drawn memory items, but with an additional consideration for the impact of

BTSP on common 1's shared among memory items. Consider a random binary memory item $\boldsymbol{x}$ with $c$ common 1's shared across overlapping memory items, and the remaining elements having an input density of $\hat{f}_p$. Unlike the case with randomly drawn memory items, here the probability of synaptic weight updates associated with these common 1's needs to be recalculated. The active state of each synaptic weight $w_{ji}$ is still determined by the parity of updates; however, the probability of these updates is altered due to the repeated occurrence of common 1's.

We now derive the conditional probability $P\left(\hat{1}(\sum_{k \in S(j)} x_{ki}) = 1 | S_j\right)$ given the size $s_j$ of $S(j)$ in two cases:

- For any $j_1 \in Q(\boldsymbol{x})$ and any $i \in supp(\boldsymbol{x})$, we estimate the probability $\hat{p}_e(s_{j_1})$ that $w_{j_1 i}$ is active given the size $s_{j_1}$ of $S(j_1)$:

$$\hat{p}_e(s_{j_1}) := P\left(\hat{1}\left(\sum_{k \in S(j_1)} x_{ki}\right) = 1 | \boldsymbol{x}, s_{j_1}\right)$$
$$= P\left(\sum_{k \in S(j_1)} x_{ki} - 1 \text{ is even } | s_{j_1}\right) = \frac{1}{2}\left(1 + (1 - \hat{f}_p)^{s_{j_1} - 1}\right).$$
$$(22)$$

- For any $j_2 \notin Q(\boldsymbol{x})$ and any $i \in supp(\boldsymbol{x})$, we estimate the probability $\hat{p}_o(s_{j_2})$ that $w_{j_2 i}$ is active given the size $s_{j_2}$ of $S(j_2)$:

$$\hat{p}_o(s_{j_2}) := P\left(\hat{1}\left(\sum_{k \in S(j_2)} x_{ki}\right) = 1 | \boldsymbol{x}, s_{j_2}\right)$$
$$= P\left(\sum_{k \in S(j_2)} x_{ki} \text{ is odd } | s_{j_2}\right) = \frac{1}{2}\left(1 - (1 - \hat{f}_p)^{s_{j_2}}\right).$$
$$(23)$$

Given the above estimates of the probability of synaptic weights remaining active after learning, we can employ similar methods as used for the randomly drawn patterns to estimate the neuron firing probability and the expected Hamming distance between different types of memory traces. The main difference in the estimation lies in modeling the effect of common bits on the neuron firing probability. We notice that the overall effect of common bits can be equivalently modeled by the parity of a variable, $\hat{s}_j$, representing the number of weight updates for a synapse connecting to neurons receiving common 1's during learning. That is to say, the active state of the weights depends on the parity of the number of weight updates: if $\hat{s}_j$ is even, the synapse will be inactive after learning, contributing zero to the weighted sum; if $\hat{s}_j$ is odd, the synaptic weight will remain active and the weighted sum can be easily modelled. By leveraging this property, we model the effects of common bits using the conditional probability and thereby provide detailed estimates of the firing probability of memory neurons after learning $M$ overlapping memory items (see Eq. S4). Then, taking the same methods for Eqs. (21) and (23), we further leverage the estimated firing probability and derive the expected HD between memory traces for the original and masked memory items (see Eq. S7), and the expected HD between memory traces for different overlapping memory items (see Eq. S8), as detailed in Supplementary Section 3.

### Details of experimental setup and simulations
**Detailed justification of the stochastic aspect of the BTSP rule of Eq. 1.** The probability 0.5 occurs in our BTSP rule because the experimental data show that if the arrival time of the input pattern $\boldsymbol{x}$ falls into the interior window from -3 to 2s relative to the onset of the plateau potential, then LTP is applied to all synaptic weights whose value is not yet maximal. If the arrival time of the input pattern falls into the outer

parts of the 10s window, i.e., into the window from -6 to -3s or from 2 to 4s relative to the onset of the plateau, then LTD is applied to all synapses whose weight is not yet minimal. Since there is no obvious dependence between the arrival times of the input pattern and the onsets of plateau potentials, we assume that the arrival time of the input pattern falls with probability 0.5 into the inner part of the 10s window, and with probability 0.5 into the outer part. Hence, with probability 0.5 a synaptic weight is not updated when an input pattern arrives within the window from −6 to 4s relative to the onset of a plateau potential because the arrival time of the input pattern falls into the "wrong" part of this 10s window where no update of its current binary weight value is possible.

**Details of simulation of continuous weight changes by BTSP rule (Fig. 1).** In Fig. 1F, continuous weight changes were simulated using Eq. (1) from ref. 3, with parameters: $w^{max} = 8$, $\tau_{ET} = 1000ms$, $\tau_{IS} = 500ms$, $k^+ = 0.5$, $k^- = 0.5$, $\alpha^+ = 0.01$, $\beta^+ = 35$, $\alpha^- = 0.01$, $\beta^- = 800$. For Fig. 1G, two neurons connected by eight synapses with binary synaptic weights were modeled. The initial binary weights were randomly sampled from a Bernoulli distribution(probability 0.5). The experiment involved 10,000 simulations, with results indicating average weight changes. In Fig. 1D and E, we established a two-layer feedforward network consisting of 400 input neurons and one output neuron and applied the binary BTSP of Eq. (1) for the connection weights. Details of the simulation setup are provided in section S2 of the Supplement.

**Details of analysis of the masking fraction for randomly drawn memory items (Fig. 2).** Unless otherwise specified, in all our experiments, we initialized the feedforward weights with a zero matrix and employed the following default parameters for modelling: $m = 25,000$, $n = 39,000$, $f_p = 0.005$, $f_q = 0.005$, $f_w = 0.6$. For the data points in the simulation, we sampled $M$ at intervals of 2000 and sampled the masking fraction at intervals of 0.02. Theoretical predictions were made for the same data points as those used in the simulations.

For every masking fraction $f_d$, memory items $x'$ were generated by randomly masking a fraction $f_d$ of the 1's in $x$. We selected the optimal $\upsilon_{th}$ by minimizing the predicted HD ratio (i.e., relative dissimilarity). Throughout the optimization process, we used a fixed masking fraction of 33% and the maximum number of input patterns $M$ (i.e., 30,000) for the predicted HD ratio by Eqs. (16) and S4. The prediction for panel C was obtained using Eq. (16),the prediction for panel D using Eqs. (13), and the prediction for panel E using Eqs. (20) and (16).

**Details of comparison with other memory models (Fig. 3).** We built the BTSP, HFN, and RP models with the same input size and connection probability. To establish the HFN with sparse binary inputs, we followed the previous studies (see Eq.5[28]) and formalized the weight matrix $W^H$ of the HFN by

$$w_{ji}^H = \begin{cases} 0, & i = j, \\ \frac{1}{m}\sum_{h=1}^{M}(x_{hi} - f_p)(x_{hj} - f_p), & i \neq j. \end{cases} \quad (24)$$

During the recall phase, we searched for the optimal threshold of the state update function (see Eq.3[28]) from [0.0, 0.02]. Referring to the memory capacity defined in section 17.2.4[39], we adopted an optimization criterion that maximizes the number of patterns that HFN can recall correctly within the error tolerance $m * 0.001$ for the dissimilarity (i.e., HD) between the memory traces of the original pattern and the masked pattern. We used the synchronous update rule and iterated the state of all neurons 100 times for retrieval.

We constructed the RP using binary connection weights. The weights were initialized with the same fraction of weights with value 1 created by BTSP after learning 30,000 patterns (see Fig. 2D for the fraction). We binarized the projection results by applying a fixed threshold, which was optimized using a grid search approach by

simulation and employing the same optimization criterion as for the BTSP.

**Details of the analysis of the masking fraction for overlapping memory items (Fig. 4).** For each input pattern, we generated common 1's corresponding to the number $c = \lfloor mf_c f_p \rfloor$. The remaining bits were then generated at a density of $\hat{f}_p = (mf_p - c)/(m - c)$, ensuring the average number of 1's per pattern matched the value $mf_p$ used for randomly drawn memory items. Corresponding data sets were generated for each specific value of $c$, and 33% of elements in each overlapping memory item were randomly masked during the recall phase. Predicted ratios were derived from Eqs.S7 and S8. We applied the grid search and optimized the threshold $\upsilon_{th}$ for overlapping memory items.

**Details of experimental setup for the input completion (Fig. 5).** For the sake of simplicity, we trained direct synaptic connections from the memory neurons to the input neurons with a simple Hebbian-like learning rule during the learning phase for the $M$ memory items. Each input pattern $x$ is presented once, as before, but we now let it persist for two steps. In the first step, random plateau signals are generated in the memory neurons, and the feedforward connections from the input to the memory neurons are adjusted via BTSP. In the second step, the feedback connection weights were adjusted via a fast Hebbian-like learning mechanism: if both the pre-and postsynaptic neuron of a feedback connection assume value 1, the synaptic weight of this feedback connection is incremented by 1; otherwise it remains unchanged. If the weight value exceeded one, it was set to a saturation value of one. These feedback connections have a density of 0.6, as for the feedforward connection. To enhance the performance in practical applications, we employed a linear function ($\upsilon_f = \alpha x + \beta$), with the average fraction of 1's as the input variable and the threshold as the output, to set thresholds for both input and memory neurons. For each masking fraction, a subset of input patterns (20%) was randomly sampled to estimate the average fraction of 1's, serving as the input for the linear functions. Specifically, for the BTSP, we established the linear function for the memory neurons with coefficients $\alpha_1 = 4707.05$ and $\beta_1 = 10.7$ and for the memory neurons, with coefficients $\alpha_2 = 692.3$ and $\beta_2 = 26.6$. These coefficients were obtained through preliminary simulation. Initially, we simulated various thresholds within the range [1, 100] for input and memory neurons for every masking fraction during the recall phase and recorded the $HD(x, r(x'))$ between each pair of thresholds. The pair of thresholds that yielded the smallest HD for each masking fraction was then utilized as data points for the function fitting.

For the binary line drawing task, we employed $50 \times 50$ pixel sparse binary line drawings as input patterns. Each pattern comprised five 8-pixel lines, randomly positioned and oriented either vertically or horizontally. In a sequence of 2000 patterns, we included pairs of similar patterns, each sharing three overlapping lines. The task parameters were set as $n = 39,000$, $f_q = 0.005$, and $f_w = 0.6$ and the linear fitting function method was also applied to set the neuron firing thresholds for both input and memory neurons.

**Details of analysis of the repulsion effect (Fig. 6).** We generated 1000 randomly drawn memory patterns with the input density $f_p$. After that, we randomly selected one pattern and generated another pattern with a 40% overlap ratio, as well as a pattern that had no overlap 1's with the chosen pattern. The generated patterns, along with other memory items, were randomly presented to the network and BTSP was applied to each pattern once. After learning, we calculated the overlapping similarity between the selected patterns and the two specific patterns separately. We also calculated the same overlapping similarity between the projection of these patterns using the random projection model. The projection results were binarized by applying the Top-K operations, ensuring that only the $\lfloor n*fq \rfloor$ neurons with the largest projection values could be activated. We set $f_q = 0.02$ in Fig. 6B and employed

other default parameters for the simulation. The results were averaged over 20 runs with different randomly initialized parameters.

## Reporting summary

Further information on research design is available in the Nature Portfolio Reporting Summary linked to this article.

## Data availability

Data generated in this paper are publicly available and are provided at https://github.com/yjwu17/simpleBTSP.

## Code availability

Source codes for reproducing the results in this paper are available at https://github.com/yjwu17/simpleBTSP, with Zenodo link https://zenodo.org/records/14133221[40].

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

## Acknowledgements

We would like to thank Jeffrey C. Magee, Aaron D. Milstein, and Sandro Romani for their helpful comments on the first version of this manuscript[9], and Alice Dauphin for a critical reading of the current version. This research was partially supported by the Human Brain Project (Grant Agreement number 785907) of the European Union, the National Science Foundation of the USA (EFRI BRAID project 2318152), the Austrian Science Fund (FWF) (10.55776/COE12), a project from the Hong Kong Polytechnic University (P0050631), and the TU Graz Open

Access Publishing Fund. For open access purposes, the author has applied a CC BY public copyright license to any author accepted manuscript version arising from this submission.

## Author contributions

Y.W. and W.M. jointly designed the learning rule and the experiments, and wrote the paper. Y.W. carried out the experiments and designed the theory.

## Competing interests

The authors declare no competing interests.
