## [Transparent Peer Review file · Nature Communications]

A simple model for Behavioral Time Scale Synaptic Plasticity provides content addressable memory with binary synapses and one-shot learning

Corresponding Author: Professor Wolfgang Maass

A version of this paper was originally rejected for publication by Nature Communications, however that decision was reconsidered after appeal by the authors.

Version 0:

Reviewer comments:

Reviewer #1

(Remarks to the Author)

Wu and Mass review

Despite the title and the language of the paper, this paper is not about BTSP. Behavioral time synaptic plasticity is primarily about the length of the time scale of the time between pre and post synaptic events that can lead to synaptic plasticity. This is on the behavioral scale of seconds rather than tens on milliseconds indicated by STDP. Another aspect related to timing is that in BTSP LTP occurs at shorter durations, while LTD at longer durations. In addition, in BTSP the postsynaptic event is not neuronal spiking but rather a plateau potential. These postsynaptic events in Hippocampus seem close to random. Another less significant aspect of STDP is the weight dependence of synaptic modifications, such weight dependence was previously observed in other forms of synaptic plasticity.

Two aspects in the paper are related to BTSP, first the random nature of the postsynaptic event and second the weight dependence of the shape of synaptic plasticity. However, in general this paper is unrelated to BTSP. This paper really about a new form of a content addressable memory, similar to the Willshaw model of 1969. It is indeed novel in some respects, as the weights are binary, and the outputs are randomly chosen and are sparse. A nice aspect of the paper is that stochastic binary weights can approximate a deterministic continuous variable. However, importantly, this is not a paper about BTSP.

Apart from this major flaw there are many incorrect statements in the text.

- A binary Hopfield model does not significantly reduce the capacity – if a batch model weight matrix is used. This is only problematic for more realistic sequential learning e.g: Fusi.
- According to Fusi and Abbott, any level of discretization reduces the scaling of the capacity significantly.
- “Orthogonality of patterns that are to be stored is an inherent requirement of HFNs” - This is not true. The original formulation Hopfield model uses random, non-orthogonal, patterns. For orthogonal inputs the capacity of the Hopfield model is N , not $0.14N$.
- Very verbose explanation of BTSP. Could be significantly shortened.
- How does equation 1 capture the timing dependence of LTP and LTD? It does not and this is the main problem with the paper, or at least with the claim that it is related to BTSP.
- Very sparse increases capacity. This was extensively studied in the 90s. For example, see Amit and Tsodyks 1991.

(Remarks on code availability)

Reviewer #3

(Remarks to the Author)

The work, "One-shot learning and robust recall with BTSP, a biological synaptic plasticity rule" describes a mathematical model for BTSP experimental observations. A simplified model was developed to match averaged observations, and the authors describe how rapid one-shot memory formation could be realized in this model, having some desirable characteristics.

A key strength of this work, which was well-described by the authors, is the connection to experimental results in mammalian studies [Milstein, et al, 2021]. The manuscript presented here attempts to incorporate numerical parameters, wherever possible, that are neuro-anatomically plausible. The non-STDP nature of the learning rule was also very intriguing.

Despite my initial excitement in the beginning of the manuscript, I ultimately had difficulty to understand what exactly was studied in their numerical models and the comparisons to other memory models (Random Projections and more classical Hopfield network). The Methods sections and Supplemental information were provided by the authors, but unfortunately were still not adequate for this reader.

One major issue I had was with the Hopfield network model comparison. In the main text, page 8, line 285, the authors note that they compared their binary weight BTSP to "continuous weight" Hopfield network. Yet, the Methods section 2.0 and Supplementary Information, Fig S7 clearly state that the Hopfield network was made with "binarized" weights, using their own developed thresholding approach. Some of the poor results for the Hopfield network were then hard to believe as these should have performed much better with continuous weights. In particular, if they actually followed the methodology of Tsodyks, M. V. and Feigel'man, M. V. (1988), which is optimized for sparse input patterns (presumably the case here, where $f_p = 0.005$ was used) this should have led to a memory capacity of approximately $N * [p * \text{abs}(\ln(p))]^{-1}$. This would be much larger than N (39000 memory neurons). Yet, for example, figures 3E and 3H suggest their Hopfield network could store less than 10000 patterns, regardless of the masking fraction.

However, even the above (done correctly, or at least more clearly) is not the most interesting comparison. Since the paper of Tsodyks and Feigelman in 1988, there have been many developments in recent years of improved Hopfield network models. Notably [2016 NeurIPS, D Krotov, et al.] and [2019 NeurIPS Chaudhuri, Fiete, et al.] describe dense associative memories and error-correcting bipartite networks, respectively, each of which exhibit memory capacities that are exponential in the number of memory neurons, or can be tuned to some polynomial in neurons ($\sim N^k$). These also exhibit few-shot learning, but do require higher order interactions between the neurons (in the case of Krotov) or a two-layer structure (in the case of Chaudhuri). The authors are correct that such network models involve continuous valued weights, and hence a binary version (not studied, to my best knowledge) could be expected to have lower capacity. These, in my opinion, would be more relevant comparison points to the present BTSP model.

Many of the features highlighted by the authors (repulsion effect, few-shot learning, robustness to masking and errors) seem to already be present in other models. It would be beneficial for the authors to perform better comparisons to state-of-the-art models and with much more clear descriptions of their methodology in these comparisons.

(Remarks on code availability)

Version 1:

Reviewer comments:

Reviewer #1

(Remarks to the Author)

Response to rebuttal:

The fact that the Magee lab calls a similar rule BTSP, in both a preprint and private communications does not convince me that a rule that does not really depend on time is indeed BTSP. I respect the Magee lab immensely, as I respect the Mass lab, which does not mean that if they think a rule is BTSP that it is indeed BTSP. If that were the case, we can do away with the review process.

When looking at the data, for example in figure 1B of this paper it is apparent that LTD occurs at larger time differences than LTP. This could be because of longer traces for LTD or due to a different non-linear transformation of a single trace, this does not fundamentally matter – the time scale is clearly different. So why do the authors claim this is still BTSP. This is

based on two factors. First, they assume binary weights, and binary one-shot plasticity. This means that they interpret the black curve of figure 1B as plasticity of synapses that are at $w_i=0$ and the red curves for a mix of synapses, some at $w_i=0$ and some at $w_i=1$, allowing for LTD as well. However, in the red curve still the time difference matters and LTD occurs primarily for a longer time difference between presynaptic activity and plateau potential. So why do they ignore it? They allow themselves to ignore the time aspect of behavioral time synaptic plasticity, because of the assumption that the plateau potentials are random, so in the formation of memories (not place fields) the timing difference between presynaptic activity and plateau potential is random. So, in the sense of the specific abstract setup of how memories are formed in this model the timing aspect is irrelevant, and can be replaced with a probability. They therefore turn this timing dependent model to a more abstract stochastic model in which time is broken into chunks of 10 seconds and a mapping from inputs to random sparse outputs is generated. This though is true only with respect to the simplified way that memories are generated in this model. Is this relevant to biology though and to the experimental findings of BTSP?

They claim in the rebuttal: "This clearly explains that our BTSP rule of Eq. 1 does capture the timing dependence of LTP and LTD that is suggested by the available experimental data." This is not strictly true, it does not capture the data, though in context of the random mapping and the random timing of plateau potentials this might not matter. The assumption that plateau potentials are random, is a decent zero's order assumption. However, we now know experimentally that this is not strictly true as in CA1 EC3 activity which triggers plateau potentials is enhanced near rewards and place field density near rewards is also increased (Grienberger and Magee, 2022). Nevertheless, it still interesting to examine the consequences of this simplest approximation.

So, what is this paper really about? It is a feedforward model of memory based on mapping from random input patterns onto random output patterns. Here both (f_p and f_q) are very sparse. This is similar to the questions asked in the Fusi, Amit and Abbott papers about capacity in discrete synapses. One example of many is the calculations of these calculations in Dubreuil et al 2014. Note that this example includes one-shot learning. So, this is not the first attempt at mathematically modeling one-shot learning. One difference is that in the previous paper it's a recurrent network, here a feedforward network with random outputs. This should not make much of a difference in the error free calculations though. If plateau potentials are indeed random, what would happen if the same (or a similar) input patters appears more than once, but each time it is mapped to another output pattern. In any case, this calculation to the best of my knowledge is new, but mathematically not that much difference than previous models.

My main problem is not with the computations, they seem valid. Indeed, the math and computation are of high quality. Basically, what the paper says is that with one shot learning and random input output mapping given very sparse input and output patterns even with binary weights the capacity of this feedforward model is quite large. Such a claim seems valid and is consistent with previous work on similar mathematical models. Neither do I have a significant problem with their interpretation of previous results. I have primarily two problems. I still think this throws away the major aspect of BTSP, which is timing. Specific timing might be irrelevant to the model setup, but it's not irrelevant in the brain. So BTSP in the title is misleading. Second, given that this is in reality a study of mapping from sparse random input patterns to sparse random output patterns, it is not very novel. It is not that these specific results have been already shown, but mathematically similar results have been published. The fact that output patterns are no longer spikes but are now plateau potentials does not really affect the math.

What is different though on a biological level between this model and most previously models is that a spike is short, it takes milliseconds. Here the plasticity time window of postsynaptic events is ~ 10 seconds. This assumption though novel might actually be problematic. When visually scanning images, we fixate at one point for about ~ 250 ms, so ~ 40 of these fixations fit into the 10 second window. Auditory processing is faster, though olfactory processing is relatively slow. Is time scale reasonable though for changing inputs driven by the movement of the animal through space, as in the BTSP experiments. Note that the Magee lab uses a track that is 200cm long. Running speeds on the track are between 10-50mm/sec. This means that the 10 second plasticity window is on the same order as the same time scale as it takes to traverse the whole track. Therefore, most CA3 cells firing over the whole track could become associated with the CA1 cell. What is being remembers here? Certainly not the location within the track, possibly the whole track. However we know experimentally, and it has also been molded using BTSP, that CA1 place cells become localized to a limited portion of the track.

In summary, I think the calculations here are valid and should be published. I personally find that the connection to BTSP is tenuous for the reasons outlined above. Additionally, the assumptions of the model are only weakly connected to memory formation in the brain. Unfortunately, it is these connection to BTSP and real memory formation that seem to make this work significant. Again, as explained above I think these connections are weak. Abstractions are necessary for mathematical calculations to be possible, but sometime the abstractions make the connections between models and the biological reality very weak.

(Remarks on code availability)

Reviewer #2

(Remarks to the Author)
Comments communicated with the editor

(Remarks on code availability)

Reviewer #3

(Remarks to the Author)

The authors have made revisions to their paper, and clarified the specific issue that the Hopfield networks compared against *did* have continuous weights rather than binary weights. This clarification in the paper and their response is helpful.

I now understand better that the authors showed unexpectedly poor behavior for the competing Hopfield networks likely due to the constraint of 60% connectivity. They chose this number, and nearly all parameters to be biologically faithful to the experimental observations for the brain regions of interest (scaled down). Without such a constraint, it should still be mentioned that a continuous-valued Hopfield network storing such sparse input patterns following [Tsodyks 1988] should achieve far higher capacity than their BTSP rule.

I also appreciate the authors reply that the specific repulsion effect they observed (Fig 6) is likely to be unique to their implementation having the LTD mechanism. In this sense, I agree that other Hopfield network models may not have this property, although it has not been specifically tested to my knowledge. Despite this particular metric, I think that the modern Hopfield networks/DAM and Bipartite expander Hopfield networks (Chaudhuri and Fiete 2019) show overall better CAM properties having exponential capacity rather than linear in the memory neurons. In addition, the Bipartite Hopfield networks from 2019 are also binary weights.

Overall, the new revised manuscript has some improvements over the previous version, particularly in clarifications on their contributions. Ultimately, I do see this work as quite interesting for the neural modeling and related communities, since it does a nice job of evaluating the experimentally observed BTSP rule, uses specific experimentally relevant parameters, and then showcases the range of interesting capabilities of the resulting CAM.

On a less positive side, however, I remain doubtful this will be of broad interest outside of these communities. In particular, the communities interested in higher performance associative memories (e.g., modern Hopfield networks), or hardware efficiency in neuromorphic computing, would unfortunately not benefit from this work compared to existing work. I must say this because the abstract ended with the sentence that they "provide a blueprint for implementing content-addressable memory with on-chip learning capability in highly energy-efficient crossbar arrays of memristors" but I think this over-states the case. In particular, the linear capacity of this network yields an extremely inefficient method to store information in a CAM. From a technological viewpoint, if one has N bits (or N neurons) of resources, the capacity from this should be 2^N . Instead, the only linear capacity of Hopfield networks has been a severe limitation for any useful hardware realization, and kept Hopfield networks more of a model theoretical system of a distributed, reconstructive memory. Practical CAMs have been built on entirely different principles using SRAM, which are also binary elements. Exciting work of the past decade developing Dense Associative Memories, Modern Hopfield Networks, and Bipartite Hopfield Networks allowed for exponential capacities, creating interest again for technology. The present work, in contrast, goes rather backwards. I stress again that I only make this statement from a technological and neuromorphic hardware perspective. I believe that the present manuscript is otherwise quite interesting, but perhaps for a narrow audience. This, ultimately, should be decided by the Nature Communications editors.

(Remarks on code availability)

Version 2:

Reviewer comments:

Reviewer #4

(Remarks to the Author)

1. My greatest question is that the classification principle of this work is similar to the STDP-based one-shot learning algorithm [1]. Please analyze the similarities and differences between this work and the algorithm of reference [1], and compare the advantages and disadvantages of the two algorithms.

[1] N. Imam, and T. A. Cleland, "Rapid online learning and robust recall in a neuromorphic olfactory circuit," Nature Machine Intelligence, vol. 2, no. 3, pp. 181-191, 2020.

2. On page 3, lines 80-82, the text states, "Since BTSP is a local synaptic plasticity rule that requires in its simplified form only two weight values, it is especially suited for on-chip learning in neuromorphic hardware, in particular for online creation and expansion of CAM incrossbar arrays of memristors with just two required resistance states."

It is good to see the BTSP algorithm only needs two binary weight values, However, I am curious to know whether the BTSP algorithm demands a greater number of synapses compared to other SNN algorithms, such as STDP.

3. Page 4 Figure 1 (b). I suggest adding necessary description about " time from plateau onset(s) " to help readers' better understanding.

4. Page 6 line 150-151 "This demonstrates that, in contrast to STDP or most other rules for synaptic plasticity, the BTSP rule of Eq. 1 is a plasticity rule for the behavioral time scale of seconds." Why can BTSP achieve one-shot learning with a shorter time window, while other rules like STDP require a longer time window?

5. Page 6 line 158-160 "We demonstrate in Fig. 1F and G that the impact of a more complex rule for BTSP with continuous weight values, such as the rule proposed in (Milstein et al., 2021), can also be modelled by the simpler rule from Eq. 1 with binary synaptic weights. "Please explain the advantages and disadvantages between the complex BTSP rules and the simpler BTSP rules proposed by the authors.

6. Page 8 line 210 " ... but so that the expected number of 1's remains the same as in the original input patterns." Why is the number of 1's the same as the original input pattern?

7. Page 8 line 214 "We measure their dissimilarity by the Hamming distance (HD) between them." This method of assessing similarity for classification appears to align with previous one-shot learning methods based on STDP [1,2]. Could you please elaborate on this? Furthermore, what motivated the authors to select HD as the measure of dissimilarity?

[1] N. Imam, and T. A. Cleland, "Rapid online learning and robust recall in a neuromorphic olfactory circuit," Nature Machine Intelligence, vol. 2, no. 3, pp. 181-191, 2020.

[2] A. Borthakur, and T. A. Cleland, "A Spike Time-Dependent Online Learning Algorithm Derived From Biological Olfaction," Frontiers in Neuroscience, vol. 13, 2019.

8. Page 8 line 232 " BTSP provides a radical departure from this postulate and also from STDP since BTSP does not depend on the firing of the postsynaptic neuron." I am interested in the specific advantages that this BTSP method offers over STDP-based rules. Please comment on it.

9. Please discuss the potential impact at large scale and future works derived from this paper.

10. It is suggested to compare and analyze the accuracy with other one-shot learning algorithms to better highlight the importance of this work.

(Remarks on code availability)

1. The code provides a README file with enough instructions for installing and running the application.

2. I am able to install and run the code.

made.

Point-by-point response to reviewer comments for NCOMMS-23-64149

Rebuttal to reviewer R1:

--The reviewer claims that this paper is not about BTSP.

The reviewer agrees with us (in a remark a few lines later) that postsynaptic events (plateau potentials) are random. In that case the BTSP rule from Eq. 1 that we study is mathematically equivalent to the BTSP rule proposed by the discoverer of BTSP (the Lab of Jeff Magee). More precisely, the study

Li, Y., Briguglio, J., Romani, S., & Magee, J. C. (2023). Mechanisms of memory storage and retrieval in hippocampal area CA3. *bioRxiv*, 2023-05.

proposes in their Eq. 1 a rule for BTSP with binary weights that is mathematically equivalent to our rule in Eq 1 in the case of random plateau potentials (the history of these rules was discussed on ll 40 of our submission; they arose from discussions between our Lab and the Lab of Jeff Magee; there are small differences regarding the lengths of time windows since their rule is based on data from area CA3).

We give in a new section S1 of the Supplement the precise formulation of this BTSP rule from (Li et al., 2023), and a proof of its equivalence to our BTSP rule for random plateau potentials. Hence the claim of the reviewer implies that also the BTSP rule from the Lab of Jeff Magee, who is the discoverer and the world's leading expert on BTSP, "is not about BTSP". This claim can obviously not be maintained.

--The reviewer asks in particular: How does equation 1 capture the timing dependence of LTP and LTD?

We had made the timing dependence of LTP and LTD in BTSP explicit in Fig. 1C, based on the experimental data of (Milstein et al., Elife 2021) that we have replotted in our Fig. 1B. One sees that the windows for LTP and LTD have about the same length of 5s. The reviewer agrees with us that in BTSP the postsynaptic event is not neuronal spiking but rather a plateau potential. These postsynaptic events in Hippocampus seem close to random. The random nature of these postsynaptic events (plateau potentials) implies that whether synaptic input falls into the LTP or LTD part of BTSP is a random decision. Since both windows have about the same length according to Fig. 1B and C, either one is applied with probability 0.5. We had explained that in ll 133 of the original submission:

Thus the dependence of the decision between LTP and LTD on the relative timing between the arrival of the synaptic input and the onset of the plateau potential can be summarized by the simple scheme of Fig. 1C. Hence, the chance that a synaptic input that arrives within the window of plasticity marked by a plateau potential falls into the LTP window is about 0.5, and the chance that it falls into the LTD window is also 0.5.

We also pointed in the lines before Eq. 1 to this fact:

When an input pattern x arrives within the time interval from -6 to 4s relative to the onset of a plateau potential, synaptic weights are increased with probability 0.5 (provided they are not yet at their maximal value of 1), and also decreased with probability 0.5 (provided they are not yet at their minimal value of 0).

This clearly explains that our BTSP rule of Eq. 1 does capture the timing dependence of LTP and LTD that is suggested by the available experimental data.

This misunderstanding by the reviewer suggests that our explanations of the BTSP rule are indeed important. Hence we are confused by the remark: “Very verbose explanations of BTSP. Could be significantly shortened.”

--In order to highlight the role of the length of the time scale of our BTSP rule we have added in new panels D and E of Fig. 1 a demonstration where we apply this rule to model the fundamental experiment by which BTSP had been discovered, see Fig.1 of (Bittner et al., Science 2017). One clearly sees that our BTSP rule captures the application of LTP to synaptic inputs that arrived several seconds before or after the onset of a plateau potential. We have also added on ll 179 the observation that the long time scale of BTSP allows us to treat in our theoretical analysis input patterns x as batch inputs, although the components of x may be dispersed over a behavioral time scale of 100s of ms or even seconds.

--The long duration of the window for plasticity in BTSP also plays a key role for an important parameter of our model: It increases the probability that synaptic input becomes subject to BTSP in a CA1 neuron. This probability is modelled by the parameter f_q in our model. We assumed for that parameter a value of 0.005, based on the estimate of

Grienberger, C., & Magee, J. C. (2022). Entorhinal cortex directs learning-related changes in CA1 representations. *Nature*, 611(7936), 554-562

that plateau potentials occur at a rate of 0.005 per second in a CA1 pyramidal cell. If a plateau potential would open the window for plasticity not for seconds, but just for 10s of ms, the probability f_q would have to be reduced by two orders of magnitude. We have shown in Fig. 6 of our submission that for such a low value of f_q one could no longer model the repulsion effect for closely related memories in the human brain (see our Fig. 6). We have also shown in Fig. S6 that the capacity of the content-addressable memory (CAM) that BTSP creates would be drastically reduced by lower values of f_q . Hence our model capitalizes in an essential way on the BTSP time scale “of seconds rather than tens of milliseconds indicated by STDP”.

--As we have explained above, our model does address the long time scale of BTSP. But this long time scale is only one of 5 salient properties of BTSP that we have listed on pages 1 and 2. We find it a bit arbitrary to single it out as the most important one.

We show in our paper, that the combination of all 5 is essential for understanding the implications of BTSP for CAM.

--The reviewer argues that the BTSP rule that we consider is “similar to the Willshaw model of 1969”. This is not correct, and inconsistent with the properties of BTSP which the reviewer had described a few lines before that claim: in BTSP the postsynaptic event is not neuronal spiking but rather a plateau potential. These postsynaptic events in Hippocampus seem close to random. In contrast, the synaptic plasticity rule of

Willshaw, D. J., Buneman, O. P., & Longuet-Higgins, H. C. (1969). Non-holographic associative memory. *Nature*, 222(5197), 960-962.

(described on the last page of that paper) is a rule where the postsynaptic event is neuronal spiking, rather than a random signal.

--In the last part of the review, R1 claims that there are many incorrect statements in the text, and lists a few points.

The first one claims that a binary Hopfield model does not significantly reduce the capacity –if a batch model weight matrix is used. But we could not find any published results which corroborate this claim, especially not for online (one-shot) learning that we consider.

Furthermore, this claim contradicts the second point of the reviewer: “according to Fusi and Abbott, any level of discretization reduces the scaling of the capacity significantly”. Apart from that, this second point does not relate to any “incorrect statement” in our paper. In fact, it is consistent with all results of our submission.

The 3rd point of the reviewer addresses the role of “orthogonality” of items that are stored in a content addressable memory. We had referred to random input patterns as being orthogonal because they are almost orthogonal. The reviewer correctly points out that they are only approximately orthogonal, and full orthogonality is a separate case. In order to make our terminology more precise we are now referring in Fig. 4 to patterns that are significantly non-orthogonal as overlapping patterns.

We had compared the performance of BTSP for overlapping input patterns in Fig. 4 C with the capacity of a HFN that arises from the standard learning rule for HFNs (Hopfield Networks). However, offline methods for setting the weights of a HFN, such as the pseudo-inverse approach that involves a matrix inversion, increase the capacity of HFNs for overlapping patterns. Since offline learning rules do not fit well into the context of our paper, we have deleted this Fig. 4C.

--The last point of the reviewer is: Very sparse increases capacity. This was extensively studied in the 90s. For example, see Amit and Tsodyks 1991. Our paper is in full agreement with this point. Hence this remark of the reviewer does not identify any “incorrect statement” of our submission. However, we have added a reference to this paper in I 299. We also want to point out that we always used the same sparsity

of input patterns, based on estimates of the sparsity of input patterns from area CA3, in all our analyses of BTSP and HFNs.

Rebuttal to reviewer R3:

R3 listed two “major issues” in the 2nd part of the review. The **first one** arises from a misunderstanding regarding the types of HFNs that we consider:

We had compared the performance of the BTSP model both with that of HFNs with continuous weights (in in Fig. 3E, 4C, 5G) and with that of HFNs with binary weights (in Fig. 5F and S7). We marked at the top of each of these panels whether the result was for HFNs with continuous or binary weights. But apparently we had confused R3 by providing in the Methods section 2.0 only further details to HFNs with binary weights (the reason was that no further details were needed for HFNs with continuous weights beyond the information that had already been provided on p.11 and 12 of the main text.). We have revised the paper so that this confusion can no longer arise. Since the comparison with HFNs with continuous weights is by far the most salient one (because the capacity of HFNs with binary weights is rather poor) , we have moved all text and figures that concern comparisons with HFNs with binary weights to section S12 of the Supplement.

The second major issue of the reviewer concerns performance results for HFNs with continuous weights:

Some of the poor results for the Hopfield network were then hard to believe as these should have performed much better with continuous weights. In particular, if they actually followed the methodology of Tsodyks, M. V. and Feigel'man, M. V. (1988), which is optimized for sparse input patterns (presumably the case here, where $f_p = 0.005$ was used) this should have led to a memory capacity of approximately $N * [p * \text{abs}(\ln(p))]^{-1}$. This would be much larger than N (39000 memory neurons). Yet, for example, figures 3E and 3H suggest their Hopfield network could store less than 10000 patterns, regardless of the masking fraction.

The reviewer had apparently overlooked that all our capacity estimates are for networks with just 60% connectivity (which we had made explicit on l. 296 of the main text). The motivation for considering networks with sparser connectivity came from data about connectivity in the hippocampus, where one does not have 100% connectivity from CA3 to CA1. We therefore focused on HFNs with the same connectivity of 60%, especially since incomplete connectivity is also hard to realize in neuromorphic hardware. In contrast, the reference Tsodyks, M. V. and Feigel'man, M. V. (1988) which the reviewer cites addresses only HFNs with 100% connectivity (that have substantially smaller capacity).

Thus, both of the “two major issue” of R3 arise from misunderstandings, and do not point to any error or defect in our results. However, we have added some words in the paper on l 297 and 352 that make these misunderstandings less likely. We have also moved all material on HFNs with binary networks into section S12 of the Supplement, so that confusion with results on HFNs with continuous weights is avoided.

Reviewer R3 raised two further issues in the last paragraph of the review. The first one is to consider also HFNs with higher order interactions among neurons (often referred to as “modern HFNs”). We have added a comment on that on ll 425 of the Discussion.

Finally, R3 claims at the end of the last paragraph, without providing any evidence or reference, the repulsion effect etc seem to be present in other models. To the best of our knowledge there exists no such reference to work where the repulsion effect was reproduced with biologically plausible rules for synaptic plasticity. Hence this claim appears to have no support. We are also not aware of other state-of-the-art models (for the topic that we consider: one-shot learning of memory traces with binary weights and local synaptic plasticity rules) to which we should add performance comparisons, as suggested in the last sentence of the review.

“One-shot learning and robust recall with BTSP, a biological synaptic plasticity rule” by Yujie Wu and Wolfgang Maass

Rebuttal to Reviewer 1 (Full text of review is cited in blue font):

The fact that the Magee lab calls a similar rule BTSP, in both a preprint and private communications does not convince me that a rule that does not really depend on time is indeed BTSP.

This claim is not correct. Our rule does depend on time. We have demonstrated this through several results in the paper, and described it more explicitly in new text on lines 144 – 151, 257 – 262, 448 - 455.

I respect the Magee lab immensely, as I respect the Mass lab, which does not mean that if they think a rule is BTSP that it is indeed BTSP. If that were the case, we can do away with the review process.

Probably the reviewer means with the informal term “is BTSP” whether a rule reflects the essential experimental data on BTSP. From the perspective of that meaning it is certainly relevant that the leading lab for experimental data on BTSP, the Magee Lab, employs for their theoretical analysis in (Li et al, 2023) a BTSP rule which is essentially equivalent to ours (see section S1 of the Supplement in our paper). It is reasonable to assume that the Magee Lab knows the essential experimental data on BTSP.

We also want to clarify that the timing aspect of BTSP is only one of several salient facets of BTSP (see our list of 5 salient facets of BTSP on lines 20 to 29, and our explanations on lines 108 to 114). Perhaps the reviewer has overlooked the other important new facets of BTSP which we describe there, e.g. that BTSP does not depend on postsynaptic firing as gating signal. Otherwise the reviewer would not have claimed that the earlier publication (Dubreuil et al., 2014) has made our theoretical analysis of BTSP obsolete: The one-shot learning rule of (Dubreuil et al., 2014) depends on postsynaptic activity as gating signal, and is therefore fundamentally different from BTSP. Perhaps the reviewer has also overlooked the important consequences of the stochasticity of the gating signal for BTSP, see our results and explanation on lines 231 – 272. In particular, we address there the fascinating relation of the stochastic aspects of BTSP to stochastic aspects of the biological algorithm from (Dasgupte et al., Science 2017).

Furthermore, we invite the reviewer to acknowledge that our BTSP rule can reproduce essential aspects of the role of time in experimental data on BTSP. We had already shown in Fig. 1 D, E of our preceding revision that our BTSP rule reproduces the main experimental result for BTSP from Fig. 1 of (Bittner et al., Science 2017): It creates in a single shot a new spatially extended place field. In our current revision we have added a further analysis of the timing aspect of BTSP in Fig. 1E. We reproduce there

for our BTSP rule the analysis that had been applied to biological data on BTSP in the landmark publication (Bittner et al., Science 2017) from the Magee Lab in order to elucidate that BTSP is fundamentally different from STDP with regard to the role of time: Its plasticity window extends over seconds, rather than over milliseconds or tens of milliseconds as for the commonly studied STDP rule. They showed in their Fig. 1 E and F that this timing aspect of BTSP has the consequence that if the mouse runs faster, the newly created place field has a longer spatial extension. We show in the new panel Fig. 1E that this fingerprint of a longer time window of plasticity also holds for our BTSP rule.

When looking at the data, for example in figure 1B of this paper it is apparent that LTD occurs at larger time differences than LTP. This could be because of longer traces for LTD or due to a different non-linear transformation of a single trace, this does not fundamentally matter – the time scale is clearly different. So why do the authors claim this is still BTSP. This is based on two factors. First, they assume binary weights, and binary one-shot plasticity. This means that they interpret the black curve of figure 1B as plasticity of synapses that are at $w_i=0$ and the red curves for a mix of synapses, some at $w_i=0$ and some at $w_i=1$, allowing for LTD as well. However, in the red curve still the time difference matters and LTD occurs primarily for a longer time difference between presynaptic activity and plateau potential. So why do they ignore it? They allow themselves to ignore the time aspect of behavioral time synaptic plasticity, because of the assumption that the plateau potentials are random, so in the formation of memories (not place fields) the timing difference between presynaptic activity and plateau potential is random. So, in the sense of the specific abstract setup of how memories are formed in this model the timing aspect is irrelevant,

This claim is not correct, see our remarks above and below.

and can be replaced with a probability. They therefore turn this timing dependent model to a more abstract stochastic model in which time is broken into chunks of 10 seconds and a mapping from inputs to random sparse outputs is generated. This though is true only with respect to the simplified way that memories are generated in this model. Is this relevant to biology though and to the experimental findings of BTSP?

The reviewer acknowledges below that the “simplified way that memories are generated in this model” is “a decent zero’s order assumption”. Furthermore, there are, to the best of our knowledge, at the moment no experimental data that suggest a modification of this assumption, except for the case of reward-based learning (which is not what we are studying in this work). In view of these facts we do not understand why the reviewer questions whether a model that is based on a “decent zero’s order assumption” is “relevant to biology and to the experimental findings of BTSP”.

They claim in the rebuttal: “This clearly explains that our BTSP rule of Eq. 1 does capture the timing dependence of LTP and LTD that is suggested by the available experimental data.” This is not strictly true, it does not capture the data, though in

context of the random mapping and the random timing of plateau potentials this might not matter. The assumption that plateau potentials are random, is a decent zero's order assumption.

We are happy that the reviewer acknowledges the fact that our model is based on a “decent zero's order assumption”.

However, we now know experimentally that this is not strictly true as in CA1 EC3 activity which triggers plateau potentials is enhanced near rewards and place field density near rewards is also increased (Grienberger and Magee, 2022).

We are not modelling reward-based learning in this paper. But a closer look at the experimental data to which the reviewer points shows that even for reward-based learning the assumption that plateau potentials are randomly distributed over time is a “decent zero's order assumption”. (Grienberger and Magee, 2022) point out in the left column on p. 558 that the increased density of CA1 place cells near reward sites can be explained primarily (precisely: by 2/3) on the basis of an invariant rate of stochastic plateau potentials. The reason is the animals spend more time near the reward site.

Nevertheless, it still interesting to examine the consequences of this simplest approximation.

So, what is this paper really about? It is a feedforward model of memory based on mapping from random input patterns onto random output patterns. Here both (f_p and f_q) are very sparse. This is similar to the questions asked in the Fusi, Amit and Abbott papers about capacity in discrete synapses.

We are examining what memory capacity can be achieved with a biologically founded rule for synaptic plasticity (BTSP). The work by Fusi, Amit and Abbott did not address this question.

One example of many is the calculations of these calculations in Dubreuil et al 2014. Note that this example includes one-shot learning. So, this is not the first attempt at mathematically modeling one-shot learning.

We have not claimed that our paper is “the first attempt at mathematically modelling one-shot learning”. Rather, we are examining functional implications of the first biologically supported rule for one-shot learning: BTSP. This rule differs in essential aspects from the one-shot learning rule (SP-model) of (Dubreuil et al., 2014) to which the reviewer points: It does not depend on postsynaptic firing. The dependence of their theoretically postulated (rather than biologically supported) rule on postsynaptic activity has the unpleasant consequence that the memory traces for older memory items are continuously weakened when further memory items are learned, see Fig. 4 a, b of (Dubreuil et al., 2014). The source for this undesirable effect is that neurons that were recruited for earlier memory traces tend to get stronger weights, and therefore have a higher probability of postsynaptic firing when new memory items are presented. This implies for the plasticity rule of (Dubreuil et al., 2014) that these neurons have a higher

probability of becoming also part of the memory trace for a subsequently presented memory item. Consequently, the fidelity of older memory traces continuously declines (see their Fig. 4 a, b). In contrast, we show in our new panel E of Fig.5 that this does not hold for BTSP: The memory traces for the first 100 in a sequence of 10.000 memory items are at the end still almost equally good as those for the last 100 memory items in this long sequence. This result demonstrates an important advance of BTSP over the previously proposed one-shot learning rule of (Dubreuil et al., 2014) to which the reviewer referred. The reason for this is that the stochastic gating signals of BTSP spread out the learning load more uniformly over the neurons, thereby combining advantages of synaptic plasticity with advantages of random projections. We explain this in detail on lines 263 to 274 and lines 469 to 474.

We are grateful to the reviewer for pointing to the previously proposed rule for one-shot learning in (Dubreuil et al., 2014). We have added this reference, and also explained on lines 375 to 380 and on lines 435 to 440 the fundamentally different properties of memory traces that are created by the rule of (Dubreuil et al., 2014) and those that are created by BTSP.

One difference is that in the previous paper it's a recurrent network, here a feedforward network with random outputs. This should not make much of a difference in the error free calculations though. If plateau potentials are indeed random, what would happen if the same (or a similar) input patters appears more than once, but each time it is mapped to another output pattern.

We are glad that the reviewer raises this question, since we had already addressed it in Fig. 6. We show there that BTSP enhances the difference between similar memory items. The LTD part of BTSP plays an essential role in that. This repulsion effect had previously been found in human memory, see e.g. the experimental data to which references are given on l. 402. Perhaps the reviewer had overlooked this important link between our BTSP model and experimental data on memory formation in the human brain

In any case, this calculation to the best of my knowledge is new, but mathematically not that much difference than previous models.

This claim that our model is “mathematically not that much difference than previous models” is wrong. The reviewer has apparently overlooked the important mechanistic and functional differences between the learning rule of (Dubreuil et al., 2014) and BTSP. In particular, the former depends on postsynaptic firing, the latter not. We have added in panel E of Fig. 4 a demonstration of the important functional impact which this difference has on the degradation of memory traces during further learning. Furthermore, we are not aware of any previous results on the formation of content addressable memories (CAMs) with a one-shot learning rule for binary synaptic weights (or with weights whose resolution stays in an experimentally supported range). Hence the claim that our models are “mathematically not that much difference than previous models” is wrong.

My main problem is not with the computations, they seem valid. Indeed, the math and computation are of high quality. Basically, what the paper says is that with one shot learning and random input output mapping given very sparse input and output patterns even with binary weights the capacity of this feedforward model is quite large. Such a claim seems valid and is consistent with previous work on similar mathematical models.

We believe that there was no previous work on similar mathematical models, see our detailed remarks above.

Neither do I have a significant problem with their interpretation of previous results. I have primarily two problems. I still think this throws away the major aspect of BTSP, which is timing. Specific timing might be irrelevant to the model setup, but it's not irrelevant in the brain. So BTSP in the title is misleading.

We are not insisting on having BTSP in the title. But the reviewer fails to provide evidence that having BTSP in the title is misleading. In fact, in order to keep notation uniform in the literature, one should note that the Magee Lab has referred to their almost identical plasticity rule in (Li et al., 2023) also as BTSP.

Furthermore, we have listed on p.1 and 2 five features in which BTSP differs from previously considered rules for synaptic plasticity, especially from STDP. One of them is the large time window for synaptic plasticity that is opened by a plateau potential. We explain on lines 144 to 151 that this enables BTSP to integrate temporally dispersed information into a single memory trace. This is demonstrated in Fig. 1 E for the case of temporally dispersed activity of presynaptic place cells. The large time window also supports the formation of conjunctive memories for other types of dispersed information, such as different frames of an episodic memory. We explain on lines 453 – 455 that a memory item in our theoretical model can be thought of an episodic memory that is dispersed over several seconds.

We also have explained on lines 257 to 262 that the long duration of the time window for BTSP is crucial for recruiting sufficiently many neurons for a memory trace, see also Fig. 2C.

Second, given that this is in reality a study of mapping from sparse random input patterns to sparse random output patterns, it is not very novel. It is not that these specific results have been already shown, but mathematically similar results have been published.

The claim that “mathematically similar results have been published” is incorrect. The only evidence which the reviewer gives is the reference to (Dubreuil et al., 2014). We have pointed out above (and on lines 375 to 380 of the paper) that their one-shot learning rule differs in essential mechanistic and functional aspects from the BTSP rule that we (and the Magee Lab) are examining.

The reviewer has also apparently not noticed the advantage of BTSP in creating content-addressable memory (CAM) with binary (or low-resolution) synapses, see Fig. 5. Similar CAM results have not been achieved with any previously published plasticity rule.

The fact that output patterns are no longer spikes but are now plateau potentials does not really affect the math.

This remark is problematic in two aspects. One is that a plateau potential is an internal signal in a neuron, not an “output pattern”. A second one is that gating plasticity by stochastic plateau potentials rather than postsynaptic firing does “affect the math.” It requires mathematical methods for analyzing stochastic, rather than deterministic plasticity rules. Furthermore, on the conceptual level it gives rise to an important new control parameter, the parameter f_q , that strongly affects the math. and network function, see Fig. 2 C-E, and lines 231 to 262. Furthermore, we have shown in Fig. 4E that this facet plays an important role in protecting memory traces from degradation during further learning. The claim that these features do “not really affect the math.” is therefore wrong.

What is different though on a biological level between this model and most previously models is that a spike is short, it takes milliseconds. Here the plasticity time window of postsynaptic events is ~10 seconds.

We agree, and we have elucidated the consequence of the length of this time window in our new Fig. 1E. In addition, we have explained on lines 257 to 262 that this length of the time window is essential (because of its impact on the parameter f_q) for creating sufficiently large memory traces (see Fig. 2C), and for reproducing the repulsion effect of human memory (see Fig. 6). We have also explained on lines 453 to 455 that it is essential for integrating temporally dispersed information into a memory trace, in particular different frames of an episodic memory.

This assumption though novel might actually be problematic. When visually scanning images, we fixate at one point for about ~250ms, so ~40 of these fixations fit into the 10 second window. Auditory processing is faster, though olfactory processing is relatively slow. Is time scale reasonable though for changing inputs driven by the movement of the animal through space, as in the BTSP experiments. Note that the Magee lab uses a track that is 200cm long. Running speeds on the track are between 10-50mm/sec. This means that the 10 second plasticity window is on the same order as the same time scale as it takes to traverse the whole track. Therefore, most CA3 cells firing over the whole track could become associated with the CA1 cell. What is being remembers here? Certainly not the location within the track, possibly the whole track. However we know experimentally, and it has also been molded using BTSP, that CA1 place cells become localized to a limited portion of the track.

We agree with the reviewer that it is non-trivial to elucidate functional advantages of the large time window of BTSP. Hence we are happy that we managed to elucidate in this work two such functional advantages, see our remarks above.

Apart from this, the review

Sugar, J., & Moser, M. B. (2019). Episodic memory: Neuronal codes for what, where, and when. *Hippocampus*, 29(12), 1190-1205

suggests why it may not be a problem that different frames of a seconds-long episode are merged into a single memory trace in area CA1: Each frame of such an episode is first combined with spatial and temporal tags (provided by neurons in EC). These tags make it possible to untangle the combined content of a memory trace.

In summary, I think the calculations here are valid and should be published. I personally find that the connection to BTSP is tenuous for the reasons outlined above. Additionally, the assumptions of the model are only weakly connected to memory formation in the brain.

The reviewer fails to point to concrete data on memory formation in the brain that are within the scope of this study, and which our BTSP rule fails to model. Hence the “weak connections to memory formation in the brain” that the reviewer claims for our model, and hence also for the BTSP model of (Li et al., 2023) of the Magee Lab, are not supported by evidence. Furthermore, the reviewer has apparently not read that part of our paper (lines 390 to 425, and Fig. 6) where we elucidate the link between BTSP and the repulsion effect of human memory. This is obviously an important and nontrivial feature of “memory formation in the brain”.

Unfortunately, it is these connection to BTSP and real memory formation that seem to make this work significant. Again, as explained above I think these connections are weak. Abstractions are necessary for mathematical calculations to be possible, but sometime the abstractions make the connections between models and the biological reality very weak.

We agree with the reviewer regarding the generic difficulty to capture essential aspects of biological reality with simple models. This generic difficulty makes it even more remarkable that our simple mathematical model for one-shot learning by BTSP reproduces the main experimental data on BTSP, and also creates functionally powerful systems of memory traces.

We also want to point out that the reviewer has not provided any evidence for the claim that the link between our BTSP model (and the almost identical model of (Li et al., 2023) from the Magee Lab) on one hand, and “biological reality” on the other hand, is “very weak”. Furthermore, the reviewer had apparently not read all parts of our paper (e.g., the section on the repulsion effect) which address this link.

Nevertheless, the fundamental criticism of the reviewer has helped us to further clarify in the revised paper some important aspects of BTSP, and of the relation of our model to other models. We are grateful for that.

.....

Rebuttal to reviewer 3 (full text of review cited in blue font)

The authors have made revisions to their paper, and clarified the specific issue that the Hopfield networks compared against *did* have continuous weights rather than binary weights. This clarification in the paper and their response is helpful.

I now understand better that the authors showed unexpectedly poor behavior for the competing Hopfield networks likely due to the constraint of 60% connectivity. They chose this number, and nearly all parameters to be biologically faithful to the experimental observations for the brain regions of interest (scaled down). Without such a constraint, it should still be mentioned that a continuous-valued Hopfield network storing such sparse input patterns following [Tsodyks 1988] should achieve far higher capacity than their BTSP rule.

We are grateful for this suggestion and have added a remark on the higher capacity of fully connected HFNs on line 313.

I also appreciate the authors reply that the specific repulsion effect they observed (Fig 6) is likely to be unique to their implementation having the LTD mechanism. In this sense, I agree that other Hopfield network models may not have this property, although it has not been specifically tested to my knowledge. Despite this particular metric, I think that the modern Hopfield networks/DAM and Bipartite expander Hopfield networks (Chaudhuri and Fiete 2019) show overall better CAM properties having exponential capacity rather than linear in the memory neurons. In addition, the Bipartite Hopfield networks from 2019 are also binary weights.

Overall, the new revised manuscript has some improvements over the previous version, particularly in clarifications on their contributions. Ultimately, I do see this work as quite interesting for the neural modeling and related communities, since it does a nice job of evaluating the experimentally observed BTSP rule, uses specific experimentally relevant parameters, and then showcases the range of interesting capabilities of the resulting CAM.

On a less positive side, however, I remain doubtful this will be of broad interest outside of these communities. In particular, the communities interested in higher performance associative memories (e.g., modern Hopfield networks), or hardware efficiency in neuromorphic computing, would unfortunately not benefit from this work compared to existing work. I must say this because the abstract ended with the sentence that they “provide a blueprint for implementing content-addressable memory with on-chip learning capability in highly energy-efficient crossbar arrays of memristors” but I think this over-states the case. In particular, the linear capacity of this network yields an extremely inefficient method to store information in a CAM. From a technological viewpoint, if one has N bits (or N neurons) of resources, the capacity from this should

be 2^N . Instead, the only linear capacity of Hopfield networks has been a severe limitation for any useful hardware realization, and kept Hopfield networks more of a model theoretical system of a distributed, reconstructive memory. Practical CAMs have been built on entirely different principles using SRAM, which are also binary elements. Exciting work of the past decade developing Dense Associative Memories, Modern Hopfield Networks, and Bipartite Hopfield Networks allowed for exponential capacities, creating interest again for technology. The present work, in contrast, goes rather backwards. I stress again that I only make this statement from a technological and neuromorphic hardware perspective. I believe that the present manuscript is otherwise quite interesting, but perhaps for a narrow audience. This, ultimately, should be decided by the Nature Communications editors.

This criticism of the reviewer is based on a misunderstanding of the state-of-the-art. In particular, the reviewer has overlooked the following 3 facts:

Fact 1: The Bipartite Hopfield networks from (Chaudhuri and Fiete, 2019) are of interest in the context of our work, since they also require just binary weights. But in contrast to our model the Bipartite Hopfield network of (Chaudhuri and Fiete 2019) **“does not model how to store externally specified inputs”**.

This fact was stated on line 542 of the more recent publication by the same senior author, Ila Fiete:

Chandra, S., Sharma, S., Chaudhuri, R., & Fiete, I. (2023). High-capacity flexible hippocampal associative and episodic memory enabled by prestructured "spatial" representations. bioRxiv, 2023-11.

Corresponding remarks can also be found in the 6th paragraph of the Discussion in (Chaudhuri and Fiete 2019). Therefore, the potential exponential capacity of Bipartite Hopfield network only holds for specially constructed sets of inputs, not for randomly generated inputs that are commonly considered for CAMs, even less for input patterns with overlap such as those that we consider in Fig. 4.

Fact 2: Other modern Hopfield networks (including Dense Associative Memories = DAMs) cannot be implemented with binary weights, or with memristors that can assume only a limited number of resistance values.

Like classical Hopfield networks, modern Hopfield networks and DAMs require that the number of weight values grows linearly with the number of patterns that are stored in the network. This fact is well-known, and obvious from the way how synaptic weights are defined in these networks, see e.g., Equ. (1) of

Bao, H., Zhang, R., & Mao, Y. (2022). The capacity of the dense associative memory networks. Neurocomputing, 469, 198-208.

Fact 3: Modern Hopfield networks and DAMs can only store more memories than the number of neurons in the network if higher order interactions between neurons (i.e., more than synaptic interactions between pairs of neurons) are allowed. Hence, they cannot be implemented with any currently available neuromorphic implementation approach.

This fact is well-known, and stated explicitly for example in the lines above equ. (9) in the standard reference

Krotov, D., & Hopfield, J. (2020). Large associative memory problem in neurobiology and machine learning. arXiv preprint arXiv:2008.06996.

This fact is stated for example also in the 5th paragraph on p. 3 of

Sharma, S., Chandra, S., & Fiete, I. (2022, June). Content addressable memory without catastrophic forgetting by heteroassociation with a fixed scaffold. In International Conference on Machine Learning (pp. 19658-19682).

The reviewer has apparently overlooked these 3 facts. As a result, the criticism of the reviewer is unfounded and invalid.

Fact 1 implies that Bipartite Hopfield networks do not provide a CAM model that could provide an alternative to our CAM model based on BTSP, since they cannot store given input patterns. They can only store input patterns that are constructed especially for this purpose.

Fact 2 implies that other modern Hopfield networks and DAMs do not provide an alternative to our model based on BTSP because they cannot be implemented with neuromorphic hardware such as memristor networks where only a limited number of weight values can be stored in the form of resistance values.

Fact 3 implies that other modern Hopfield networks and DAMs do not provide an alternative to our model based on BTSP because they cannot be implemented with any neuromorphic hardware that is based on pairwise interactions (synapses) between neurons. To the best of our knowledge, all presently known neuromorphic design technology is based on pairwise interactions between neurons.

The reviewer mentions also SRAMs as being superior to our CAM model. This may well be the case, but SRAMs are not considered to provide a path towards energy efficient neuromorphic hardware. In particular, their memory can only be maintained as long as electrical energy is provided to the network.

The reviewer states “I stress again that I only make this statement from a technological and neuromorphic hardware perspective.” But unfortunately, the reviewer is not

sufficiently informed about the state-of-the-art in this area. In particular the reviewer was apparently not aware of the 3 facts stated above.

Hence, the conclusion of the reviewer that our work is “backwards” and would only be of interest for a “narrow audience” was based on a misunderstanding of the state-of-the-art. As a result, these conclusions of the reviewer are unfounded and wrong. On the contrary, the CAM model via BTSP that this work proposes is likely to advance the state-of-the-art in neuromorphic CAM in a significant way. In addition, it provides a new functionally attractive paradigm for on-chip local learning in neuromorphic hardware.

We concede that the less advantageous aspects of modern HFNs, in particular the 3 facts above, are less prominently described in the literature. Hence other readers could become subject to a similar misunderstanding as the reviewer. Therefore we have rewritten the section on the state-of-the-art of modern HFNs and their implications for neuromorphic CAM on lines 488 – 496 in the Discussion. In particular, we have made the 3 facts that are mentioned above explicit. We are grateful to the reviewer for motivating us to do this.

On the side we would like to mention that work is underway at IBM in Zurich to implement our CAM model with BTSP on their HERMES chip with phase-change memory. Hence our work does provide a “blueprint” for a new generation of CAMs in neuromorphic hardware.

A point-by-point response to the reviewers' comments

Reviewer #4 (Remarks to the Author):

We would like to thank the reviewer very much for the insightful comments, and the suggestions to compare with further preceding work, and to highlight the importance of the results.

My greatest question is that the classification principle of this work is similar to the STDP-based one-shot learning algorithm [1]. Please analyze the similarities and differences between this work and the algorithm of reference [1] and compare the advantages and disadvantages of the two algorithms.

We have added on l 491 the following comments in response to this suggestion:

“Another biological mechanism for one-shot creation of memory traces in a biological organism had been described in (Imam, 2020) for the mammalian olfactory bulb. This work has in common with our approach that it also provides one-shot insertion of memory items into a neural network that supports recall with noisy cues. Furthermore, they consider a spiking neural network implementation. Spike-based implementations of HFNs, where the timing of spikes conveys analog information, had already been introduced in (Maass et al., 1997). In principle this model also allowed one-shot learning, because the standard learning rule for HFNs is a one-shot learning rule. But both this HFN model and (Imam 2020) requires a large number of weight values, whereas our BTSP-based model requires only binary weight values. Another difference to (Imam 2020) is that in their approach new neurons have to be added to the networks when new memory items are to be learnt. In contrast, our memory network can have a fixed architecture.

Additionally, in terms of learning rules, two different mechanisms are employed in (Imam, 2020): A simplified STDP rule for synaptic weights and a less standard rule for modifying the blocking period for firing that is caused by a spike of the inhibitory GC neuron in the postsynaptic MC cell. Since they add new GC neurons whenever a new memory item is to be learnt, their model is likely to avoid the degradation of earlier memory traces that typically occurs with Hebbian-like plasticity rules for one-shot learning, see the explanations in the text for Fig. 5. In contrast, the stochastic nature of gating signals for BTSP, in conjunction with sparse coding, reduces the chance that previously learnt memory traces are overwritten. “

Maass, W., & Natschläger, T. (1997). Networks of spiking neurons can emulate arbitrary Hopfield nets in temporal coding. Network: Computation in Neural Systems, 8(4), 355-371.

[1] N. Imam, and T. A. Cleland, “Rapid online learning and robust recall in a neuromorphic olfactory circuit,” *Nature Machine Intelligence*, vol. 2, no. 3, pp. 181-

191, 2020.

2. On page 3, lines 80-82, the text states, "Since BTSP is a local synaptic plasticity rule that requires in its simplified form only two weight values, it is especially suited for on-chip learning in neuromorphic hardware, in particular for online creation and expansion of CAM in crossbar arrays of memristors with just two required resistance states."

It is good to see the BTSP algorithm only needs two binary weight values, However, I am curious to know whether the BTSP algorithm demands a greater number of synapses compared to other SNN algorithms, such as STDP.

We have shown in Fig. S8 that the input completion capability of a network that is trained by BTSP (with binary weights) is roughly in the same range as that of a HFN with the same number of synapses with continuous weights that is trained with a Hebbian learning rule. We are not aware of capacity estimates of networks that are trained by STDP. But if the resulting capacity would be roughly the same as for HFNs, the answer to the question would be that BTSP does not require a greater number of synapses.

3. Page 4 Figure 1 (b). I suggest adding necessary description about " time from plateau onset(s) " to help readers' better understanding.

We have expanded this phrase to

"time from the onset of the gating signal (plateau potential)"

4. Page 6 line 150-151 "This demonstrates that, in contrast to STDP or most other rules for synaptic plasticity, the BTSP rule of Eq. 1 is a plasticity rule for the behavioral time scale of seconds." Why can BTSP achieve one-shot learning with a shorter time window, while other rules like STDP require a longer time window?

Apparently, the reviewer refers here to the length of time required for learning? If this interpretation is correct, one can say that BTSP requires less time for learning, because it operates with a higher learning rate. It is not clear what would happen if one would increase the learning rate for STDP in a similar manner. Possibly its dependence on spike times, which tend to be noisy, would encumber its learning performance.

5. Page 6 line 158-160 "We demonstrate in Fig. 1F and G that the impact of a more complex rule for BTSP with continuous weight values, such as the rule proposed in (Milstein et al., 2021), can also be modelled by the simpler rule from Eq. 1 with binary synaptic weights. "Please explain the advantages and disadvantages between the complex BTSP rules and the simpler BTSP rules proposed by the authors.

The simpler BTSP rule has the advantage that it is theoretically more tractable. Also, neuromorphic implementations of binary weights require less effort than for example memristors that can acquire many weight values.

We have added this explanation after line 171

6. Page 8 line 210 " ... but so that the expected number of 1's remains the same as in the original input patterns." Why is the number of '1's the same as the original input pattern?

This sentence was meant to be a description of the experimental design for Fig. S4, not as a claim or fact. We have tried to make the corresponding sentence on I 223 clearer by saying:

"We show in Fig. S4 that recall with such "two-sided" errors in the cue is in general equally successful as recall with "one-sided" errors in the cue, i.e., where only 1's are changed into 0's."

7. Page 8 line 214 "We measure their dissimilarity by the Hamming distance (HD) between them." This method of assessing similarity for classification appears to align with previous one-shot learning methods based on STDP [1,2]. Could you please elaborate on this? Furthermore, what motivated the authors to select HD as the measure of dissimilarity?

We edited the sentence on I 226 so that it reads:

"Similarly as in (Borthakur 2019) we used the Hamming distance (HD) to measure their dissimilarity, since this is arguably the simplest and most direct measure for the distance between binary vectors."

[1] N. Imam, and T. A. Cleland, "Rapid online learning and robust recall in a neuromorphic olfactory circuit," *Nature Machine Intelligence*, vol. 2, no. 3, pp. 181-191, 2020.

[2] A. Borthakur, and T. A. Cleland, "A Spike Time-Dependent Online Learning Algorithm Derived From Biological Olfaction," *Frontiers in Neuroscience*, vol. 13, 2019.

8. Page 8 line 232 " BTSP provides a radical departure from this postulate and also from STDP since BTSP does not depend on the firing of the postsynaptic neuron." I am interested in the specific advantages that this BTSP method offers over STDP-based rules. Please comment on it.

We added on I 251 the explanation:

“This has the advantage that it better protects previously formed memory traces when new memories are subsequently learnt, since it has no bias towards reusing neurons whose synaptic weights were increased when they became part of a previously formed memory trace (see Fig. \ref{Fig5: input completion}E). Another difference to STDP is that BTSP supports the separation of memory traces for very similar memory items through the repulsion effect, see Fig. \ref{Fig6: BTSP-repulsion}.”

9. Please discuss the potential impact at large scale and future works derived from this paper.

We have explained on l 329 that our analysis (see Fig. S7) suggests that the memory capacity of our model can be increased to 800k if one scales the network size up to that of corresponding areas in the human brain.

10. It is suggested to compare and analyze the accuracy with other one-shot learning algorithms to better highlight the importance of this work.

Our BTSP rule requires only binary weights. If one reduces HFN to binary weights, their performance drops drastically. This has been theoretically predicted (Tarkov, 2019) and confirmed by our numerical experiments (Fig. S9). We have added a remark on that in l 512 – 515.

Apart from (Chaudhuri and Fiete 2019) we are not aware of competing methods for storing large numbers of memories with binary weights. (Chaudhuri and Fiete 2019) offers much larger memory capacity, but this larger capacity can only be realized for specially constructed memory patterns. This is already discussed on l 524, and we would rather not add further statements to avoid redundancy.

We have rewritten the last sentence of the Discussion on l 532 to highlight the importance of the work.